# Design-Based Anytime-Valid Inference for
# Randomized Experiments with Delayed Outcomes and Staggered Entry

**Michael Lindon** [1]   **Nathan Kallus** [2]

## Abstract

Delayed outcomes are ubiquitous in online experimentation: treatment can affect whether an outcome occurs, when it occurs, and its realized value. To accommodate staggered entry while remaining robust to environmental nonstationarity and unit-level heterogeneity, we adopt a design-based perspective and target the sample cumulative reward in each arm as a function of calendar time. Our confidence sequences allow practitioners to continuously monitor the counterfactual incremental reward, such as revenue, that would have been realized by calendar time $t$ had all entered units been assigned to treatment rather than control. The main technical challenge is the choice of design-based filtration, complicated by the presence of asynchronous potential outcome times. We show that the IPW treatment-effect estimation error is not a martingale with respect to any filtration, while each arm-specific IPW estimation error is a martingale with respect to a carefully chosen arm-specific event-time filtration. We therefore construct a confidence sequence for the treatment effect by combining two arm-level confidence sequences with a union bound, and further demonstrate that this can outperform the traditional design-based variance upper bound. Finally, we characterize the class of augmentations for which the per-arm AIPW estimation error remains a martingale.

## 1. Introduction

In many randomized experiments, the outcome of interest is not realized immediately after treatment assignment, but is observed after some delay. Crucially, the treatment can affect the *value* of the outcome, the *timing* of its realization, and the *proportion* of units that ever realize an event at all (with $t_i(w) = \infty$ denoting units for which no event ever occurs). In conversion experiments, for example, the treatment may affect (i) the conversion rate, (ii) the time to conversion, and (iii) the value of the conversion (e.g. revenue). Similarly, in marketing A/B tests comparing ad campaigns, the treatment may *accelerate* the outcome arrival, sometimes referred to in marketing as a "pull-forward" effect (Neslin et al., 1985; Ailawadi et al., 2007; Gupta, 1988). Yet, while treatment outcomes arrive sooner, they may be of lesser value, or fewer in eventuality, presenting a trade-off between treatment and control.

In this work, we aim to enable continuous monitoring of experiments with delayed outcomes using anytime-valid inference methods. At any analysis time $t$, the data is *right-censored*: the outcomes for some units have not yet been observed. Crucially, the treatment itself influences this censoring. Not only are unit outcomes random, but the order in which units are observed is also random.

Our contributions are as follows. To remain robust to temporal heterogeneity and nonstationarity, we formulate delayed-outcome experimentation with staggered entry in a design-based framework where both outcome time and value are unknown fixed potential outcomes. As treatment assignment is the only source of randomness, this enables us to provide confidence sequences under a minimal set of assumptions. We target the sample cumulative reward as a function of time, a causal estimand that avoids modeling the unobserved future or assuming a superpopulation. We characterize the augmented estimators that preserve a martingale structure under a specific design-based filtration: the augmentation must activate at the event time and remain constant thereafter. We then prove a negative result: the treatment effect estimation error is not a martingale under *any* filtration. This impossibility result arises from the dependence between arms induced by randomized assignment and the temporal misalignment caused by differentially delayed outcomes. We resolve this via a union bound over single-arm confidence sequences—a necessity analogous to the variance upper bound in classical design-based inference, where cross-arm dependence cannot be estimated from ob-

---
[1]Netflix, Los Gatos, CA, USA [2]Cornell University, Ithaca, NY, USA. Correspondence to: Michael Lindon <michael.s.lindon@gmail.com>.

*Proceedings of the 43^{rd} International Conference on Machine Learning*, Seoul, South Korea. PMLR 306, 2026. Copyright 2026 by the author(s).

served data. We demonstrate that this union bound actually yields *tighter* intervals than the variance upper bound when treatment induces asymmetry in outcome arrival rates.

**Code Availability.** A reference implementation of the proposed method is available at `https://github.com/michaellindon/icml2026`.

**Conflict of Interest Disclosure.** Michael Lindon is employed by Netflix, and Appendix C analyzes Netflix software telemetry data. The authors are not aware of any other financial conflicts of interest related to this work.

## 2. Background

### 2.1. Design-Based vs. Superpopulation

Randomization (design-based) inference treats potential outcomes as fixed and averages only over the treatment assignment (Fisher, 1935; Splawa-Neyman et al., 1990; Cochran, 1977; Rosenbaum, 2002; Rubin, 1974; Imbens & Rubin, 2015; Freedman, 2008). By contrast, superpopulation analyses additionally posit that the observed experimental units are draws from an underlying distribution and target population-average parameters (Abadie et al., 2020). Both perspectives are widely used; the differences between the estimand (sample vs. population) and the uncertainty quantification (randomization vs. sampling) can be subtle (Ding, 2017).

Our focus is the design-based perspective because online experiments are often entangled with time-varying behavior. Units entering the experiment through time, as opposed to being randomly sampled simultaneously from a superpopulation, are fundamentally heterogeneous. Seasonality, promotions, and new product releases introduce nonstationarity in the observed outcomes. The heterogeneity mix of units combined with the nonstationarity of the environment make the joint superpopulation distribution of outcome times and values hard to justify. In view of this, uncensored observations from an earlier-admitted unit cannot be used to infer anything about a censored outcome of a later-admitted unit, ruling out the use of survival-analytical solutions. Related discussions emphasize that superpopulation assumptions are typically unverifiable and can be fragile under interference, heterogeneity, and non-random sampling (Freedman, 2008). Conversely, the design-based perspective *does not* make assumptions about the joint distribution of outcome times and values, and only assumes the potential outcomes exist. Our estimand is the *sample cumulative reward* (and its treatment-control difference) as a function of calendar time, i.e., what would have been observed up to time $t$ had all enrolled units been assigned to a given arm. This aligns with common product metrics such as cumulative revenue or conversions, and crucially, avoids extrapolating beyond the currently observed censoring horizon, focusing instead on the counterfactual of having used just one or the other campaign to this point.

### 2.2. Anytime-Valid Inference

Continuous monitoring invalidates classical fixed-horizon $p$-values and confidence intervals due to optional stopping. A large literature in sequential analysis addresses this via time-uniform guarantees, including confidence sequences (Wald, 1947; Darling & Robbins, 1967; Robbins, 1970; Robbins & Siegmund, 1970; Howard et al., 2021) and, more recently, game-theoretic/$e$-value approaches (Shafer, 2021; Ramdas et al., 2023; Grünwald et al., 2024).

Most anytime-valid constructions assume a *fixed* sequence (or order of units) with *random* outcomes. Differentially delayed outcomes break this approach because treatment can change *which* units have a realized outcome by time $t$, so the sequence itself becomes treatment-dependent. For this reason we provide confidence sequences with respect to an external clock $t$ instead of an enumeration over units. Closest to our setting are (i) design-based confidence sequences for randomized experiments without delay (Ham et al., 2022), (ii) proportional hazards-based sequential logrank tests (Gu & Lai, 1991; ter Schure et al., 2024), and (iii) anytime-valid inference for counting processes (Lindon & Malek, 2022; Lindon & Kallus, 2025). We contribute to this related-work landscape by developing design-based, time-uniform inference for cumulative reward processes under delayed outcomes and staggered entry, avoiding proportional hazards assumptions and targeting a calendar-time estimand that does not require modeling unobserved tails.

In the presence of nonstationarity, the purpose of anytime-valid inference is not to stop the experiment early and declare an optimal treatment arm. Instead it concerns providing *honest* inference about what is known *so far*, without contradicting ourselves in the future. Indeed, even with fully observed outcomes, which arm is better may depend on how long one waits, given that the fraction or value of conversions may well be inversely related to how soon they are realized. Declaring an arm optimal at any point $t$ would require making very strong assumptions about the kind of nonstationarity that can occur in the future. We provide this inference to the researcher as the experiment progresses, and defer to them to take the appropriate action (if any). In many cases, this may involve continuously monitoring the experiment until all units are observed for a pre-specified window.

## 3. Observations, Estimands, and Estimators

To formalize our design-based approach to delayed outcomes, we consider the time of realization itself as a poten-

*Table 1.* First five rows of Dataset 5.1. Staggered entry times $E_i$, potential time and outcome values $t_i(w)$ and $y_i(w)$ for $w \in \{0, 1\}$, treatment assignment $w_i$, probability of treatment $\pi_i(1)$, and observed time $t_i$ and outcome $y_i$. A hyphen indicates that the corresponding event never occurs, so the outcome value does not apply.

| $i$ | $E_i$ | $t_i(0)$ | $t_i(1)$ | $y_i(0)$ | $y_i(1)$ | $w_i$ | $\pi_i(1)$ | $t_i$ | $y_i$ |
|---|---|---|---|---|---|---|---|---|---|
| 0 | .0506 | $\infty$ | 1.29 | – | 0.79 | 0 | 0.5 | $\infty$ | – |
| 1 | .0552 | 19.49 | 1.48 | 1.20 | 0.69 | 1 | 0.5 | 1.48 | 0.69 |
| 2 | .0695 | $\infty$ | 1.00 | – | 0.75 | 1 | 0.5 | 1.00 | 0.75 |
| 3 | .0920 | 19.53 | $\infty$ | 1.43 | – | 1 | 0.5 | $\infty$ | – |
| 4 | .1084 | 9.46 | $\infty$ | 1.59 | – | 0 | 0.5 | 9.46 | 1.59 |

tial outcome (see Table 1), as interventions can affect both the timing and the value of the outcome.

Let $E_i$ denote the entry time of unit $i$ and assume units are enumerated in increasing order of entry time. Note, however, that the order in which unit *outcomes* are *observed* depends upon the realized treatment assignments. Let $(t_i(w), y_i(w))$ denote the potential time and outcome of unit $i$ under treatment $w \in \{0, 1\}$. We define the *potential reward process* for unit $i$ under treatment $w \in \{0, 1\}$ as

$$r_{it}(w) = y_i(w)\mathbf{1}[t_i(w) \leq t], \qquad (1)$$

where $\mathbf{1}[t_i(w) \leq t]$ denotes the indicator function. This formulation models a single delayed event per unit. Let $\mathcal{E}_t := \{i : E_i \leq t\}$ be the index set of units who have entered the experiment by time $t$, and let $N_t := |\mathcal{E}_t|$ denote the number of entered units by time $t$. We further sum over all units who have entered the experiment by time $t$ to define the total reward process under treatment $w$ by time $t$:

$$r_t(w) = \sum_{i \in \mathcal{E}_t} r_{it}(w). \qquad (2)$$

This defines an estimand of primary interest – how much revenue would have been generated as a function of time if all units were assigned to treatment $w$. We can then define the difference to determine the incremental reward, as a function of time, provided by the treatment:

$$\Delta_t := r_t(1) - r_t(0). \qquad (3)$$

Notice our estimand is not a property of a distribution, it directly targets the business metric of interest (e.g. incremental revenue) and circumvents the need for distributional assumptions as the only source of randomness is via treatment assignment. We further remark that this estimand is in fact already implicit in common industry practice: when analyzing an experiment at time $t$, practitioners routinely impute the outcome as zero for any unit whose event has not yet been observed. Although often regarded as an ad-hoc convenience, this is precisely the $y_i(w)\mathbf{1}[t_i(w) \leq t]$ term in equation (1).

We assume a randomized experiment in which units are independently assigned to arm $w$ with probability $\pi_i(w) =$

$\mathbb{P}[W_i = w]$, where $\pi_i(w) > \pi_{\min} > 0$. All expectations and variances throughout are taken with respect to this randomization distribution, treating potential outcomes as fixed. Under standard SUTVA assumptions (no interference, no multiple versions of treatment), we can write the observed quantities for unit $i$ in terms of the potential outcomes through the switching equations:

$$y_i = w_i y_i(1) + (1 - w_i) y_i(0), \qquad (4)$$
$$t_i = w_i t_i(1) + (1 - w_i) t_i(0), \qquad (5)$$
$$r_{it} = w_i r_{it}(1) + (1 - w_i) r_{it}(0), \qquad (6)$$

denoting the observed outcome, observed time, and observed revenue process for unit $i$. We assume the potential outcomes are bounded: $|y_i(w)| \leq B$ for all $i$ and $w \in \{0, 1\}$.

We assume access to a vector of pre-treatment covariates $X_i$ for each unit, and let $m_{it}(w) = m(X_i, w, t)$ denote a pointwise prediction of $r_{it}(w)$. The augmented inverse probability weighted (AIPW) estimator is:

$$\hat{r}_{it}(w) = m_{it}(w) + \frac{\mathbf{1}[w_i = w]}{\pi_i(w)} e_{it}(w), \qquad (7)$$

where $e_{it}(w) = r_{it}(w) - m_{it}(w)$ is the residual of the working model (Robins et al., 1994). Setting $m_{it}(w) = 0$ recovers the Horvitz-Thompson (IPW) estimator (Horvitz & Thompson, 1952). Under randomization, we have

$$\mathbb{E}[\hat{r}_{it}(w)] = r_{it}(w), \qquad (8)$$
$$\mathbb{V}[\hat{r}_{it}(w)] = \frac{1 - \pi_i(w)}{\pi_i(w)} e_{it}(w)^2. \qquad (9)$$

The unbiasedness holds for any choice of $m_{it}(w)$, while the variance depends on the squared residual $e_{it}(w)^2$ it induces. Similarly defining $\hat{\Delta}_{it} = \hat{r}_{it}(1) - \hat{r}_{it}(0)$ and $\Delta_{it} = r_{it}(1) - r_{it}(0)$, we have

$$\mathbb{E}[\hat{\Delta}_{it}] = \Delta_{it}, \qquad (10)$$
$$\mathbb{V}[\hat{\Delta}_{it}] = \frac{1 - \pi_i(1)}{\pi_i(1)} e_{it}(1)^2 + \frac{1 - \pi_i(0)}{\pi_i(0)} e_{it}(0)^2$$
$$+ 2 e_{it}(1) e_{it}(0) \qquad (11)$$
$$\leq \sigma_{it}^2 := \frac{e_{it}(1)^2}{\pi_i(1)} + \frac{e_{it}(0)^2}{\pi_i(0)}. \qquad (12)$$

The last inequality is the traditional design-based upper bound on the variance, $\sigma_{it}^2$, commonly used to compute conservative *pointwise* confidence intervals in the classical setting (Splawa-Neyman et al., 1990; Imbens & Rubin, 2015). It is necessary because we cannot estimate the cross term, since we do not simultaneously observe both potential outcomes. As we describe later in Remark 4.11, an analogous phenomenon manifests in the sequential setting,

whereby the dependence between arms precludes the existence of a common filtration under which both arms are martingales.

An unbiased estimator of the upper bound on the variance is:

$$\hat{\sigma}_{it}^2 = \hat{e}_{it}(1)^2 + \hat{e}_{it}(0)^2, \tag{13}$$

where $\hat{e}_{it}(w) = \hat{r}_{it}(w) - m_{it}(w) = \frac{\mathbf{1}[w_i=w]}{\pi_i(w)} e_{it}(w)$ is the IPW-weighted residual. We similarly define our estimators of the cumulative and incremental reward processes as

$$\hat{r}_t(w) = \sum_{i \in \mathcal{E}_t} \hat{r}_{it}(w), \quad \text{and} \quad \hat{\Delta}_t = \hat{r}_t(1) - \hat{r}_t(0). \tag{14}$$

# 4. Confidence Sequences for $\hat{r}_t(w)$ and $\hat{\Delta}_t$

Our goal in this section is to construct asymptotic confidence sequences for $r_t(w)$ and $\Delta_t$. The route is through strong approximation (Strassen, 1964; 1967): under the regularity conditions of Lemma 4.5 and an appropriate martingale structure, the estimation error process can be coupled to a time-changed Brownian motion. Inverting the normal-mixture boundary for this Brownian approximation then yields an asymptotic confidence sequence (Waudby-Smith et al., 2024). The central issue is therefore to choose a filtration under which the estimation error is both adapted and has zero conditional drift.

In the design-based framework, we condition on the realized entry times, covariates, and potential outcomes. Conditional on these quantities, the only random variables are the treatment assignments. Define the centered assignment weight

$$Z_i(w) := \frac{\mathbf{1}[w_i = w]}{\pi_i(w)} - 1.$$

For a fixed arm $w$, the unit-level estimation error is

$$M_{it}(w) := \hat{r}_{it}(w) - r_{it}(w) = Z_i(w) e_{it}(w), \tag{15}$$

and the cumulative estimation error is

$$M_t(w) := \hat{r}_t(w) - r_t(w) = \sum_{i \in \mathcal{E}_t} M_{it}(w). \tag{16}$$

This decomposition isolates the filtration problem. The potential reward path $r_{it}(w)$ is fixed by conditioning, while a candidate augmentation path must be fixed in advance or measurable from information available before it is used. Thus the new randomness in the unit-$i$ jump enters through $Z_i(w)$. If the filtration exposes $Z_i(w)$ before the residual can jump, the jump is generally predictable; if it exposes $Z_i(w)$ after the residual has jumped, the process is not adapted. We therefore compare three candidate filtrations. The conditioned-on quantities appearing in these filtrations record calendar-time availability; they are not additional sources of randomness.

**Definition 4.1** (Designer Filtration). Let $\mathcal{G}_t = \sigma(\{(E_i, X_i, w_i) : E_i \leq t\})$. This corresponds to the filtration where the treatment assignment $w_i$ is revealed at the entry time $E_i$.

**Definition 4.2** (Observer Filtration). Let $\mathcal{H}_t := \sigma(\{(E_i, X_i) : E_i \leq t\} \cup \{w_i : t_i \leq t\})$. This corresponds to the filtration where the treatment assignment $w_i$ is revealed at the observation time $t_i$.

**Definition 4.3** (Single-Arm Filtration). For each arm $w$, let $\mathcal{F}_t(w) := \sigma(\{(E_i, X_i) : E_i \leq t\} \cup \{w_i : t_i(w) \leq t\})$. This corresponds to the $w$-specific filtration where the treatment assignment $w_i$ is revealed at the potential event time $t_i(w)$.

The designer filtration is natural from the experimenter's perspective: users are randomized when they enter, so the assignment is immediately available to the experiment designer. For the IPW estimator this makes the arm-$w$ error process adapted, but it reveals the assignment too early for a martingale argument. Consider the IPW special case $m_{it}(w) = 0$. At the instant before the arm-$w$ event, $\mathcal{G}_{t_i(w)-}$ already contains $w_i$, while $y_i(w)$ and $t_i(w)$ are fixed by conditioning. The unit-level jump is therefore predictable:

$$\begin{aligned} \mathbb{E}\big[M_{i,t_i(w)}(w) &- M_{i,t_i(w)-}(w) \mid \mathcal{G}_{t_i(w)-}\big] \\ &= Z_i(w) y_i(w). \end{aligned} \tag{17}$$

This conditional expectation is generally nonzero, so the unit-level martingale decomposition already fails under $\mathcal{G}_t$.

The observer filtration has the opposite problem. It reveals $w_i$ only when the observed outcome is realized at $t_i = t_i(w_i)$. This is the realized event time, whereas the arm-$w$ error process jumps at the potential event time $t_i(w)$. Consequently, $M_t(w)$ need not be adapted to $\mathcal{H}_t$. For example, in the IPW special case, suppose $t_i(1) < t_i(0)$ and $w_i = 0$. For any $s$ such that $t_i(1) < s < t_i(0)$, the arm-1 error has already jumped and equals

$$M_{is}(1) = \left( \frac{\mathbf{1}[w_i = 1]}{\pi_i(1)} - 1 \right) y_i(1),$$

whose value depends on $w_i$. But $w_i$ is not yet part of $\mathcal{H}_s$ because the observed event has not occurred. Thus the unit-level arm-1 error is not $\mathcal{H}_s$-measurable.

The single-arm filtration is aligned with the arm-$w$ jump time. It hides $w_i$ before $t_i(w)$ and reveals it exactly when the arm-$w$ potential event occurs. This is not generally an observed-data filtration, because it depends on the counterfactual event time $t_i(w)$ for units assigned to the other arm. It is nevertheless the filtration aligned with the design-based randomization needed for single-arm martingale structure: among the three candidates above, it is the only one that reveals assignment at the same time as the corresponding arm-specific jump.

## 4.1. Single-Arm Inference

We now characterize when the unit-level arm-$w$ error process is a martingale under $\mathcal{F}_t(w)$. The condition is unitwise. At time $t_i(w)$, the filtration reveals $w_i$; immediately before that time, $w_i$ is still unrevealed. Thus any augmentation value multiplied by $Z_i(w)$ at the jump must already be determined from the strict past. For nontrivial augmentations, we therefore require $m_{i,t_i(w)}(w)$ to be $\mathcal{F}_{t_i(w)-}(w)$-measurable. This rules out using unit $i$'s own treatment assignment, but permits functions of pre-treatment covariates and of outcomes or assignments available strictly before $t_i(w)$.

**Theorem 4.4** (Unit-Level Single-Arm Martingale Condition). *Fix an arm $w$ and a unit $i$, and let $M_{it}(w)$ be as in equation (15). Suppose $m_{i,t_i(w)}(w)$ and $\pi_i(w)$ are $\mathcal{F}_{t_i(w)-}(w)$-measurable, with $\mathbb{P}(w_i = w \mid \mathcal{F}_{t_i(w)-}(w)) = \pi_i(w)$. Then $(M_{it}(w))_t$ is adapted to $\mathcal{F}_t(w)$ and has zero conditional mean increments if and only if*

$$m_{it}(w) = 0 \qquad \text{for all } t < t_i(w),$$
$$m_{it}(w) = m_{i,t_i(w)}(w) \quad \text{for all } t \geq t_i(w).$$

*Equivalently, since the increments are bounded, $(M_{it}(w))_t$ is a martingale with respect to $\mathcal{F}_t(w)$.*

Under the conditions of Theorem 4.4, all unit-level processes are martingales with respect to the same filtration, so linearity of conditional expectation implies that the cumulative estimation error process

$$M_t(w) = \sum_{i \in \mathcal{E}_t} M_{it}(w)$$

is also a martingale with respect to $\mathcal{F}_t(w)$. Theorem 4.4 shows that the martingale requirement is highly restrictive. For each unit, the only martingale-preserving augmentations are event-time augmentations,

$$m_{it}(w) = f_i(w)\mathbf{1}[t_i(w) \leq t],$$

where $f_i(w)$ is $\mathcal{F}_{t_i(w)-}(w)$-measurable. Thus the augmentation must be zero before the arm-$w$ potential event time and constant afterward. This characterization is structural but generally oracle: for a fixed arm $w$, computing the activation indicator requires knowing $t_i(w)$ for every unit, whereas randomized assignment reveals $t_i(w)$ only for units assigned to arm $w$. Thus nontrivial event-time AIPW estimators are not generally constructible from the realized data. The IPW estimator remains the directly practical special case ($f_i(w) = 0$), while event-time AIPW serves as an oracle target for what practical AIPW procedures should approximate.

To see the martingale structure, order the units by their potential event times $t_i(w)$ such that $t_{\sigma(1)}(w) \leq t_{\sigma(2)}(w) \leq \cdots$. The process $M_t(w)$ changes value only at the discrete event times $(t_{\sigma(i)}(w))_{i \in \mathbb{N}}$. At each jump, $Z_i(w)$ is revealed for the first time under $\mathcal{F}_t(w)$, while the residual $e_{i,t_i(w)}(w)$ is measurable from the strict past by Theorem 4.4. Thus each jump has conditional mean zero. Let

$$V_t(w) := \sum_{i \in \mathcal{E}_t} \frac{1 - \pi_i(w)}{\pi_i(w)} e_{it}(w)^2, \qquad (18)$$

$$\hat{V}_t(w) := \sum_{i \in \mathcal{E}_t} (1 - \pi_i(w))\hat{e}_{it}(w)^2, \qquad (19)$$

denote the predictable quadratic variation and its plug-in estimator. Under the event-time condition, $e_{it}(w) = 0$ before $t_i(w)$ and is constant thereafter, so $V_t(w)$ increases only at arm-$w$ potential event times. The estimator $\hat{V}_t(w)$ is adapted to $\mathcal{F}_t(w)$ under the event-time condition. It is also estimable from the observed arm-$w$ samples: random assignment reveals $(t_i(w), y_i(w))$ for units with $w_i = w$, and these observed residual jumps provide an unbiased sample for the squared-jump clock.

Since $(M_t(w))_t$ is a square-integrable martingale with bounded increments, we can approximate it via a strong invariance principle.

**Lemma 4.5** (Strassen (1964; 1967)). *Fix an arm $w \in \{0, 1\}$ and suppose the event-time condition of Theorem 4.4 holds. Assume the arm-$w$ residual jumps are uniformly bounded, propensities are bounded away from zero, and $V_t(w) \to \infty$. Then, on a potentially enriched probability space, there exist independent standard Gaussians $(G_i)_{i \geq 1}$ such that:*

$$\hat{r}_t(w) - r_t(w) =$$
$$\sum_{i:t_i(w) \leq t} \nu_i G_i + o_{a.s.}\left(V_t(w)^{3/8} \log V_t(w)\right), \qquad (20)$$

*where $\nu_i^2 = \frac{1 - \pi_i(w)}{\pi_i(w)} e_{i,t_i(w)}(w)^2$.*

Such strong approximations are now standard in developing asymptotic anytime-valid procedures (Waudby-Smith et al., 2024; Bibaut et al., 2022; Ham et al., 2022).

Following Waudby-Smith et al. (2024), we say that intervals $(\hat{\theta}_t \pm L_t)_t$ form a $(1 - \alpha)$-asymptotic confidence sequence for $(\theta_t)_t$ if they approximate some nonasymptotic $(1 - \alpha)$-confidence sequence with bounds $L_t^*$ such that $L_t^*/L_t \to 1$ almost surely. Leveraging the strong approximation and a mixture martingale provided in the appendix, we establish the following result for any estimator satisfying the event-time condition of Theorem 4.4.

**Theorem 4.6** (Asymptotic Confidence Sequence for $r_t(w)$). *Assume outcomes are bounded ($|y_i(w)| \leq B$), propensities are bounded away from zero ($\pi_i(w) \geq \pi_{\min} > 0$), and $V_t(w) \to \infty$ as $t \to \infty$. For any $\eta^2 > 0$, define the boundary for the error process $\hat{r}_t(w) - r_t(w)$ as*

$$b(V; \alpha) = \sqrt{\frac{V\eta^2 + 1}{\eta^2} \log\left(\frac{V\eta^2 + 1}{\alpha^2}\right)}. \qquad (21)$$

*Then $\hat{r}_t(w) \pm b(\hat{V}_t(w); \alpha)$ is a $(1-\alpha)$ asymptotic confidence process for $r_t(w)$.*

The boundary $b(V; \alpha)$ is the normal-mixture boundary from Howard et al. (2021) and is used similarly in Waudby-Smith et al. (2024); Ham et al. (2022). Note that the asymptotic definition above requires only that the ratio $L_t^*/L_t \to 1$, not that the intervals shrink. The intervals need not shrink on the cumulative scale, because $r_t(w)$ itself grows with $t$. If desired, one can divide both arm-level intervals by $N_t$ to obtain intervals for average cumulative reward. Under bounded residuals and growing $N_t$, these rescaled intervals tighten with time. This common rescaling does not change whether the two arm-level intervals overlap at a fixed time. We therefore work on the cumulative scale, which matches the objective of comparing total rewards.

### 4.2. Multiple-Arm Inference

Under the event-time condition of Theorem 4.4, the arm-level errors are martingales, but with respect to different filtrations: $(\hat{r}_t(1) - r_t(1))_t$ is a martingale under $\mathcal{F}_t(1)$, while $(\hat{r}_t(0) - r_t(0))_t$ is a martingale under $\mathcal{F}_t(0)$. The treatment-effect error

$$\hat{\Delta}_t - \Delta_t = (\hat{r}_t(1) - r_t(1)) - (\hat{r}_t(0) - r_t(0))$$

therefore cannot be handled by simply subtracting two martingales under a common filtration. The following covariance argument shows a stronger obstruction: generically, $\hat{\Delta}_t - \Delta_t$ is not a martingale under any filtration.

**Lemma 4.7.** *Let $(X_t)_{t \geq 0}$ be a square-integrable martingale with respect to some filtration, and suppose $X_0 = 0$. Then, for any $0 \leq s \leq t$,*

$$\text{Cov}(X_s, X_t) = \text{Var}(X_s). \tag{22}$$

**Proposition 4.8.** *The covariance of the error process is*

$$\text{Cov}(\hat{\Delta}_s - \Delta_s, \hat{\Delta}_t - \Delta_t) = \text{Cov}(\hat{\Delta}_s, \hat{\Delta}_t)$$
$$= \sum_{i \in \mathcal{E}_t} \mathbb{E}[\hat{\Delta}_{is}\hat{\Delta}_{it}] - \Delta_{is}\Delta_{it}$$
$$= \sum_{i \in \mathcal{E}_t} \left\{ \frac{e_{is}(1)e_{it}(1)}{\pi_i(1)} + \frac{e_{is}(0)e_{it}(0)}{\pi_i(0)} \right.$$
$$\left. -(e_{is}(1) - e_{is}(0))(e_{it}(1) - e_{it}(0)) \right\}, \tag{23}$$

*while the variance is*

$$\text{Var}(\hat{\Delta}_s - \Delta_s) = \sum_{i \in \mathcal{E}_s} \left\{ \frac{e_{is}(1)^2}{\pi_i(1)} + \frac{e_{is}(0)^2}{\pi_i(0)} \right.$$
$$\left. -(e_{is}(1) - e_{is}(0))^2 \right\}. \tag{24}$$

By the contrapositive of Lemma 4.7, if there exist $s < t$ such that

$$\text{Cov}(X_s, X_t) \neq \text{Var}(X_s),$$

then the process $(X_t)_t$ cannot be a martingale with respect to any filtration. Proposition 4.8 shows that this covariance identity fails for $\hat{\Delta}_t - \Delta_t$ in general. Thus the treatment-effect error is not, in general, a martingale under any filtration. The martingale "violation" $\text{Cov}(\hat{\Delta}_s, \hat{\Delta}_t) - \text{Var}(\hat{\Delta}_{\min(s,t)})$ for Dataset 5.1 is visualized in Figure 5 (Appendix B). The same obstruction applies to both IPW and oracle event-time AIPW, except in degenerate cases where the residual products cancel.

Since the treatment-effect error is not itself a martingale, we construct valid inference arm by arm. Applying Theorem 4.6 separately to $r_t(1)$ and $r_t(0)$ and then taking a union bound yields a confidence sequence for $\Delta_t$.

**Theorem 4.9** (Asymptotic Confidence Sequence for $\Delta_t$). *Under the assumptions of Theorem 4.6 for both $w = 0$ and $w = 1$, let $b(V; \alpha)$ be defined as in that theorem. Then $\hat{\Delta}_t \pm \left( b(\hat{V}_t(0); \alpha/2) + b(\hat{V}_t(1); \alpha/2) \right)$ is a $(1 - \alpha)$ asymptotic confidence process for $\Delta_t$.*

The proof follows by applying Theorem 4.6 separately to each arm at level $\alpha/2$ and taking a union bound.

**Corollary 4.10** (Asymptotic Sequential P-Value). *Consider testing the null hypothesis $H_0 : \Delta_t = 0$ (no treatment effect at time $t$). The sequential p-value is defined as*

$$p_t = \inf\{\alpha \in (0, 1] : |\hat{\Delta}_t| > b(\hat{V}_t(0); \alpha/2)$$
$$+ b(\hat{V}_t(1); \alpha/2)\}, \tag{25}$$

*with the convention that $p_t = 1$ if the set is empty. When the set is nonempty and the infimum is attained in $(0, 1)$, $p_t$ equivalently solves*

$$|\hat{\Delta}_t| = b(\hat{V}_t(0); p_t/2) + b(\hat{V}_t(1); p_t/2). \tag{26}$$

*Remark* 4.11 (Cross-Arm Dependence in Fixed-Sample and Sequential Settings). The union bound approach mirrors a familiar phenomenon in fixed-sample design-based inference. In the classical setting, the variance of $\hat{\Delta}_{it}$ decomposes as

$$\text{Var}(\hat{\Delta}_{it}) = \text{Var}(\hat{r}_{it}(1)) + \text{Var}(\hat{r}_{it}(0))$$
$$- 2\text{Cov}(\hat{r}_{it}(1), \hat{r}_{it}(0)), \tag{27}$$

where $\text{Cov}(\hat{r}_{it}(1), \hat{r}_{it}(0)) = -e_{it}(1)e_{it}(0)$ arises from the perfect negative correlation between $\mathbf{1}[w_i = 1]$ and $\mathbf{1}[w_i = 0]$. This covariance term involves both potential outcomes and is therefore unestimable, forcing practitioners to use the variance upper bound $\sigma_{it}^2$ which drops it.

In the sequential setting, the same cross-arm dependence manifests differently: the covariance structure $\text{Cov}(\hat{\Delta}_s, \hat{\Delta}_t) \neq \text{Var}(\hat{\Delta}_s)$ for $s < t$ (Proposition 4.8) violates the martingale property under any filtration. Both

phenomena stem from a single root cause: treatment assignment induces dependence between arm estimators that cannot be jointly characterized. In fixed-sample inference, this dependence renders a variance component unestimable; in sequential inference, it precludes the existence of a common filtration under which both arm processes are martingales.

The union bound is thus the sequential analog of the variance upper bound: both treat each arm separately, where valid inference is available, and combine the results conservatively.

### 4.3. Union Bound vs Variance Upper Bound

The union-bound construction may appear conservative because it avoids a direct boundary for $\hat{\Delta}_t - \Delta_t$. However, the natural direct variance for this error process is not estimable even at a fixed time. As described in Section 3, the exact design-based variance in (11) contains the cross-potential residual product $e_{it}(1)e_{it}(0)$, while random assignment reveals only one potential residual per unit. Classical practice therefore uses the upper bound $\sigma_{it}^2$ in (12). Here we compare our valid sequential boundary to the corresponding cumulative upper-bound benchmark. For constant assignment probability $\pi_i(1) = \pi$, write

$$\sigma_t^2 := \sum_{i \in \mathcal{E}_t} \sigma_{it}^2$$
$$= \sum_{i \in \mathcal{E}_t} \left\{ \frac{e_{it}(1)^2}{\pi} + \frac{e_{it}(0)^2}{1-\pi} \right\} = \frac{V_t(1)}{1-\pi} + \frac{V_t(0)}{\pi}. \quad (28)$$

Because Proposition 4.8 shows that $\hat{\Delta}_t - \Delta_t$ is not generally a martingale, $b(\sigma_t^2; \alpha)$ should be read as a benchmark rather than a valid confidence-sequence boundary. The next proposition shows that, asymptotically, the valid union-bound boundary need not be wider than this benchmark, and can be narrower when the arm-level variance clocks are imbalanced.

**Proposition 4.12** (Relative Width: Union vs. Variance Upper Bound). *Let $\alpha \in (0,1)$, $\eta > 0$, and let the treatment assignment probability be $\pi_i(1) = \pi \in (0,1)$ for all $i$. Define the generalized relative width function $R_\pi(V_0, V_1)$ as:*

$$R_\pi(V_0, V_1) = \frac{b(V_0; \alpha/2) + b(V_1; \alpha/2)}{b\left(\frac{V_1}{1-\pi} + \frac{V_0}{\pi}; \alpha\right)} \quad (29)$$

*If $V_0 + V_1 \to \infty$ and $V_0/(V_0 + V_1) \to \rho \in [0,1]$, then*

$$R_\pi(V_0, V_1) \to \frac{\sqrt{\rho} + \sqrt{1-\rho}}{\sqrt{\rho/\pi + (1-\rho)/(1-\pi)}} \leq 1. \quad (30)$$

*Consequently, $R_\pi(V, V) \to 2\sqrt{\pi(1-\pi)}$ in the symmetric case, $R_\pi(V_0, V_1) \to \sqrt{1-\pi}$ when $V_0 = o(V_1)$, and $R_\pi(V_0, V_1) \to \sqrt{\pi}$ when $V_1 = o(V_0)$.*

An additional benefit of the union bound is that one can report confidence sequences for each arm separately, in addition to the difference, without the need for further multiple testing correction, as it is already present. In Figure 2 in Appendix B, we illustrate the tightness of the boundary $b(V_t(0); \alpha/2) + b(V_t(1); \alpha/2)$ with respect to a hypothetical boundary $b(2(V_t(0) + V_t(1)); \alpha)$ (which equals $b(\sigma_t^2; \alpha)$ under balanced assignment) for comparison.

## 5. Simulation Study

We evaluate the empirical coverage of our confidence processes using synthetic data that mimics the structure of our motivating application. The data-generating process is designed to capture a scenario where treatment accelerates outcome realization but reduces outcome magnitude.

**Dataset 5.1** (Synthetic Nonstationary Delayed Outcomes). We generate $N = 500$ units with staggered entry times $E_i \overset{iid}{\sim} \text{Uniform}(0, 10)$ and balanced assignment $\pi_i(1) = 0.5$. The data-generating process is designed to have four features:

1. **Arm-specific never-event fractions.** With probabilities $(\theta_0, \theta_1) = (0.20, 0.35)$, a unit has $t_i(w) = \infty$ under arm $w$, so the corresponding reward does not contribute to the cumulative process.

2. **Treatment pull-forward.** Conditional on not being a never-event unit, $t_i(w)$ is generated by thinning from

$$\lambda(t \mid E, w) = \lambda_{\text{base}}(t - E, w)\lambda_{\text{cycle}}(t)\lambda_{\text{shock}}(t, w).$$

 The baseline hazard is log-normal with $(\mu_0, \sigma_0) = (2.5, 0.5)$ for control and $(\mu_1, \sigma_1) = (0.3, 0.3)$ for treatment, so treatment events typically arrive earlier.

3. **Calendar-time nonstationarity.** The hazard includes the cyclical term

$$\lambda_{\text{cycle}}(t) = 1 + 0.2\sin(2\pi t/7)$$

 and localized shocks

$$\lambda_{\text{shock}}(t, w) = 1 + \sum_j a_{jw}\exp\{-(t - c_j)^2/(2\sigma_j^2)\}.$$

 The shocks are centered at $c_1 = 8$ and $c_2 = 15$, with arm-specific intensities $(a_{1,0}, a_{1,1}) = (1.5, 2.0)$ and $(a_{2,0}, a_{2,1}) = (1.0, 0.8)$.

4. **Outcome-time dependence.** For finite event times, rewards are

$$y_i(w) = \beta_w\{1 + 0.15\log(1 + t_i(w) - E_i)\}$$
$$\times \{1 + 0.1\sin(2\pi t_i(w)/7)\}\epsilon_i,$$

 where $(\beta_0, \beta_1) = (1.0, 0.6)$ and $\epsilon_i \sim \text{Uniform}(0.9, 1.1)$.

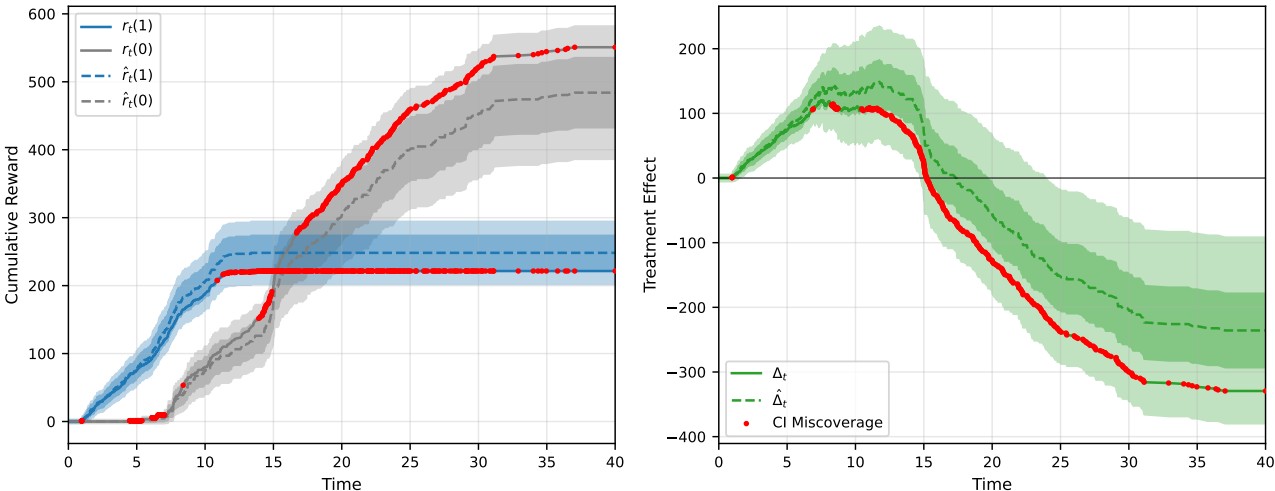

*Figure 1.* A single realization from Dataset 5.1 comparing classical pointwise confidence intervals (dark shaded) from equation (31) with the anytime-valid confidence processes from Theorems 4.6 and 4.9 (light shaded). Left panel: cumulative rewards $r_t(w)$ and estimates $\hat{r}_t(w)$ for control (gray) and treatment (blue), showing the "pull-forward" effect where treatment outcomes arrive earlier but control accumulates higher reward over time. Right panel: treatment effect $\Delta_t$ and estimate $\hat{\Delta}_t$. Red dots indicate times where the pointwise interval fails to cover the truth.

Together, these features create a setting where treatment appears better early because outcomes arrive sooner, while control eventually wins in cumulative reward because its outcomes have larger magnitude; see Figure 3 in Appendix B for an illustration.

We restrict the main simulation study to IPW, the directly observable estimator covered by Theorem 4.6. Entry-frozen AIPW augmentations can be attractive for variance reduction, but their anytime-valid analysis requires controlling the error introduced by activating the augmentation at entry rather than at the unit's potential event time. This extra term is not covered by the martingale argument in Section 4. We leave such extensions to future work.

Figure 1 provides a comparison of classical pointwise confidence intervals:

$$\hat{r}_t(w) \pm \sqrt{\chi^2_{1,1-\alpha}\hat{V}_t(w)} \quad \text{and} \quad \hat{\Delta}_t \pm \sqrt{\chi^2_{1,1-\alpha}\hat{\sigma}^2_t} \quad (31)$$

with our IPW confidence sequences. The red dots show clear coverage violations for the classical pointwise confidence intervals, whereas the IPW confidence sequences cover the estimand at all times.

We examine empirical time-uniform coverage of the IPW confidence sequences by repeatedly drawing treatment assignments while holding the potential outcomes fixed. For each replication, we evaluate whether $r_t(w)$ and $\Delta_t$ are contained in their confidence sequences for all monitoring times $t$. Across 200 treatment-assignment resamples, the empirical time-uniform coverage of the IPW confidence se-

quences was 96.0% for $r_t(0)$, 97.0% for $r_t(1)$, and 99.0% for $\Delta_t$, compared with nominal anytime coverage levels of 97.5% for each arm and 95% for the treatment effect.

Appendix C applies the IPW construction to telemetry data from a software A/B "canary" test monitoring out-of-memory (OOM) errors. OOM errors are a natural delayed-outcome setting: memory leaks may take minutes or hours to produce a crash, depending on device state and usage patterns. With $y_i(w) = 1$, the cumulative reward estimand becomes a counterfactual counting process, so the confidence sequence tracks the difference in cumulative OOM counts between treatment and control while preserving time-uniform error control. Equivalently, this is a design-based analogue of monitoring counterfactual cumulative incidence curves, with randomization rather than a survival model providing the basis for inference.

## 6. Conclusion

Delayed-outcome experiments are difficult because treatment can affect more than a single scalar outcome. It can change the probability that an event ever occurs, the delay until that event occurs, and the value realized when it occurs. Rather than choosing one of these dimensions as the outcome, we collapse them into the sample cumulative reward process $r_t(w)$: the total reward that would have been observed by calendar time $t$ had all entered units been assigned to arm $w$. This gives a time-indexed causal estimand that directly represents what experimenters monitor in practice.

Staggered entry introduces an additional layer of complexity. Units entering the experiment at different times may differ systematically, so units separated by entry time are often heterogeneous. At the same time, external factors such as product changes, seasonality, or economic conditions may affect outcome timing and value, introducing nonstationarity in the outcome distribution. Together, these features make a stable superpopulation model for delayed outcomes hard to justify. For this reason, we adopt a design-based framework and target a sample causal estimand: we condition on the entry times, covariates, and potential outcomes, leaving only the treatment assignments random.

Our main technical contribution is to show that, in this design-based setting, the treatment-effect estimation error is not a martingale with respect to any filtration. This obstruction is the sequential analogue of the classical fixed-sample variance problem: the exact design-based variance contains an unobservable cross-potential-outcome term, so classical practice replaces it with a variance upper bound. Sequentially, the same cross-arm dependence prevents a direct martingale confidence sequence for $\Delta_t$. We therefore construct confidence sequences arm by arm, where valid martingale inference is possible, and combine them using a union bound.

For a single arm, the appropriate filtration reveals unit $i$'s treatment indicator at its arm-$w$ potential event time $t_i(w)$. Under this filtration, the IPW estimation error is a martingale, yielding design-based confidence sequences for arm-level cumulative rewards. The same analysis also characterizes the oracle event-time AIPW estimators that preserve the martingale property: nonzero augmentations must activate at the potential event time and remain fixed thereafter. The IPW estimator is the directly observable special case covered by the main theory.

The union-bound construction is not merely a convenience; it plays the same role sequentially that the variance upper bound plays at fixed sample sizes. It treats the arms separately, where valid inference is available, and then combines the two arm-level confidence sequences into a confidence sequence for $\Delta_t$. Perhaps surprisingly, this need not be a severe price: asymptotically, the union-bound boundary can be no wider, and often narrower, than the usual variance-upper-bound benchmark when the arm-level variance clocks are imbalanced.

The resulting method provides anytime-valid IPW inference for cumulative rewards under staggered entry and delayed outcomes without distributional assumptions on event times or rewards. Simulations and the telemetry case study illustrate the practical value of this perspective: the method tracks the evolving counterfactual reward processes while maintaining time-uniform error control. The main open direction is variance reduction. Although AIPW estimators are

fixed-time unbiased under randomization, the martingale-preserving augmentations identified by our theory require oracle event-time information: they must activate at $t_i(w)$ for every unit, but randomized assignment reveals $t_i(w)$ only for units assigned to arm $w$. Thus the only directly observable estimator covered by the present anytime-valid theory is IPW. Developing confidence sequences that leverage predictions of both $y_i(w)$ and $\mathbf{1}[t_i(w) \leq t]$ to reduce uncertainty is the focus of future work.

## Impact Statement

This paper presents work whose goal is to advance the field of Machine Learning. There are many potential societal consequences of our work, none of which we feel must be specifically highlighted here.

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

# A. Proofs

## A.1. Proof of Theorem 4.4

*Proof.* We prove both directions for the unit-level process.

**If direction.** Assume $m_{it}(w) = 0$ for all $t < t_i(w)$ and $m_{it}(w) = m_{i,t_i(w)}(w)$ for all $t \geq t_i(w)$.

We first verify adaptedness. For $t < t_i(w)$, the residual is $e_{it}(w) = r_{it}(w) - m_{it}(w) = 0 - 0 = 0$, so $M_{it}(w) = 0$ regardless of the unobserved $w_i$. For $t \geq t_i(w)$, the treatment assignment $w_i$ is $\mathcal{F}_t(w)$-measurable, and $m_{i,t_i(w)}(w)$ is $\mathcal{F}_{t_i(w)-}(w)$-measurable by assumption, hence $\mathcal{F}_t(w)$-measurable. Therefore $e_{it}(w) = y_i(w) - m_{i,t_i(w)}(w)$ and $M_{it}(w) = Z_i(w)e_{it}(w)$ are $\mathcal{F}_t(w)$-measurable.

Next, we establish the zero conditional mean increment property. Since $m_{it}(w)$ is constant on $[t_i(w), \infty)$, the process $(M_{it}(w))_t$ changes value only at $t_i(w)$. The increment at $t_i(w)$ is:

$$\Delta M_{i,t_i(w)}(w) = Z_i(w) \cdot (y_i(w) - m_{i,t_i(w)}(w)). \tag{32}$$

According to the filtration, the treatment assignment $w_i$ is not revealed until time $t_i(w)$. By the predictability and correct specification of the propensity,

$$\mathbb{E}[Z_i(w) \mid \mathcal{F}_{t_i(w)-}(w)] = \frac{\mathbb{P}(w_i = w \mid \mathcal{F}_{t_i(w)-}(w))}{\pi_i(w)} - 1 = 0. \tag{33}$$

Since $y_i(w) - m_{i,t_i(w)}(w)$ is $\mathcal{F}_{t_i(w)-}(w)$-measurable, it factors out:

$$\mathbb{E}[\Delta M_{i,t_i(w)}(w) \mid \mathcal{F}_{t_i(w)-}(w)] = (y_i(w) - m_{i,t_i(w)}(w)) \cdot \mathbb{E}[Z_i(w) \mid \mathcal{F}_{t_i(w)-}(w)] = 0. \tag{34}$$

As the expected increment is zero at the only jump time and the process is constant otherwise, $(M_{it}(w))_{t \geq 0}$ has zero conditional mean increments.

**Only if direction.** We show that violating either condition prevents adaptedness or the zero conditional mean increment property.

*Case 1: $m_{it}(w) \neq 0$ for some $t < t_i(w)$.* At such a time, $e_{it}(w) = -m_{it}(w) \neq 0$ (since $r_{it}(w) = 0$). Since $t < t_i(w)$, the treatment assignment $w_i$ is not $\mathcal{F}_t(w)$-measurable, so $M_{it}(w) = Z_i(w)e_{it}(w)$ is not $\mathcal{F}_t(w)$-measurable. The process fails to be adapted.

*Case 2: $m_{it}(w)$ is not constant on $[t_i(w), \infty)$.* Suppose $m_{it}(w) = 0$ for $t < t_i(w)$ and the post-event values $m_{it}(w)$ are $\mathcal{F}_t(w)$-measurable, so the process is adapted, but there exist $t_i(w) \leq s < t$ such that $m_{is}(w) \neq m_{it}(w)$. Over this interval, $r_{it}(w) = r_{is}(w) = y_i(w)$, so
$$M_{it}(w) - M_{is}(w) = Z_i(w)\{m_{is}(w) - m_{it}(w)\}. \tag{35}$$

Since $w_i$ is already $\mathcal{F}_s(w)$-measurable, this is a predictable nonzero change, so its conditional expectation given $\mathcal{F}_s(w)$ equals the increment itself rather than zero. This violates the zero conditional mean increment property. $\qquad\square$

## A.2. Proof of Theorem 4.6

*Proof.* The proof follows standard arguments for asymptotic confidence sequences under non-identical variances (Waudby-Smith et al., 2024; Ham et al., 2022), which we include for the sake of completeness. The boundedness assumption ensures the Lindeberg condition is satisfied.

**Step 1: Strong approximation.** Let $\nu_i^2 := \frac{1-\pi_i(w)}{\pi_i(w)} e_{i,t_i(w)}(w)^2$ denote the variance contribution from unit $i$, so the cumulative variance is $V_t(w) = \sum_{i:t_i(w) \leq t} \nu_i^2$. By Lemma 4.5, on a potentially enriched probability space there exist i.i.d. standard Gaussians $(G_i)_{i \geq 1}$ such that

$$\hat{r}_t(w) - r_t(w) = \sum_{i:t_i(w) \leq t} \nu_i G_i + o_{a.s.}\left( (V_t(w) \log \log V_t(w))^{1/2} \right). \tag{36}$$

**Step 2: Gaussian mixture martingale for non-identical variances.** For any $\lambda \in \mathbb{R}$, the process

$$\widetilde{M}_t(\lambda) := \exp\left\{ \sum_{i:t_i(w) \leq t} \left( \lambda \nu_i G_i - \frac{\lambda^2 \nu_i^2}{2} \right) \right\} \tag{37}$$

is a nonnegative martingale starting at one. Mixing over $\lambda$ with a mean-zero Gaussian distribution with variance $\eta^2 > 0$, we obtain the closed-form mixture martingale

$$\widetilde{M}_t := \int_{\lambda \in \mathbb{R}} \widetilde{M}_t(\lambda) \frac{1}{\sqrt{2\pi\eta^2}} \exp\left\{ -\frac{\lambda^2}{2\eta^2} \right\} d\lambda = \frac{\exp\left\{ \frac{\eta^2 \left( \sum_{i:t_i(w) \leq t} \nu_i G_i \right)^2}{2(V_t(w)\eta^2 + 1)} \right\}}{\sqrt{V_t(w)\eta^2 + 1}}, \tag{38}$$

which is also a nonnegative martingale with $\widetilde{M}_0 = 1$.

**Step 3: Applying Ville's inequality and variance estimation.** By Ville's inequality (Ville, 1939), $\mathbb{P}(\exists t : \widetilde{M}_t \geq 1/\alpha) \leq \alpha$. Inverting this bound, with probability at least $(1 - \alpha)$,

$$\forall t > 0, \quad \left| \sum_{i:t_i(w) \leq t} \nu_i G_i \right| < \sqrt{\frac{V_t(w)\eta^2 + 1}{\eta^2} \log\left( \frac{V_t(w)\eta^2 + 1}{\alpha^2} \right)} = b(V_t(w); \alpha). \tag{39}$$

Combining with Step 1, we have with probability at least $(1 - \alpha)$,

$$\forall t > 0, \quad |\hat{r}_t(w) - r_t(w)| < b(V_t(w); \alpha) + o_{a.s.}\left( (V_t(w) \log\log V_t(w))^{1/2} \right). \tag{40}$$

It remains to show that $\hat{V}_t(w)$ can replace $V_t(w)$. For the unaugmented IPW estimator, the potential outcomes are fixed constants, so taking the unconditional expectation over treatment assignments yields $\mathbb{E}[\hat{V}_t(w)] = V_t(w)$. However, when using an event-time augmentation like the running mean, the residuals $e_{it}(w)$ depend on past assignments, making the true predictable quadratic variation $V_t(w)$ a random process. Because the augmentation $m_{it}(w)$ is strictly $\mathcal{F}_{t_i(w)-}(w)$-measurable, the residual $e_{it}(w)$ is deterministic conditional on the strict past. Therefore, the expected increment of the variance estimator conditional on the past exactly matches the increment of the true variance process: $\mathbb{E}[\hat{V}_t(w) - \hat{V}_{t-}(w) \mid \mathcal{F}_{t-}(w)] = V_t(w) - V_{t-}(w)$. Because both the past estimator $\hat{V}_{t-}(w)$ and the true variance target $V_t(w)$ are $\mathcal{F}_{t-}(w)$-measurable, rearranging this yields $\mathbb{E}[\hat{V}_t(w) - V_t(w) \mid \mathcal{F}_{t-}(w)] = \hat{V}_{t-}(w) - V_{t-}(w)$, fulfilling the strict definition of a martingale for the difference $\hat{V}_t(w) - V_t(w)$. Since outcomes are bounded ($|y_i(w)| \leq B$) and propensities are bounded away from zero, the increments of this martingale are uniformly bounded. By the Strong Law of Large Numbers for Martingales, $\frac{\hat{V}_t(w) - V_t(w)}{V_t(w)} \xrightarrow{a.s.} 0$ as $V_t(w) \to \infty$, which implies $\hat{V}_t(w)/V_t(w) \xrightarrow{a.s.} 1$.

**Step 4: Showing that $\hat{V}_t(w)$ can replace $V_t(w)$.** We now show that $b(V_t(w); \alpha) = b(\hat{V}_t(w); \alpha)(1 + o(1))$. Writing $\hat{V}_t(w) - V_t(w) = o(V_t(w))$, we have $V_t(w) = \hat{V}_t(w) + o(\hat{V}_t(w))$. Substituting into the boundary:

$$b(V_t(w); \alpha)^2 = \frac{V_t(w)\eta^2 + 1}{\eta^2} \log\left( \frac{V_t(w)\eta^2 + 1}{\alpha^2} \right)$$

$$= \frac{(\hat{V}_t(w) + o(\hat{V}_t(w)))\eta^2 + 1}{\eta^2} \log\left( \frac{(\hat{V}_t(w) + o(\hat{V}_t(w)))\eta^2 + 1}{\alpha^2} \right)$$

$$= \frac{\hat{V}_t(w)\eta^2 + 1 + o(\hat{V}_t(w))}{\eta^2} \log\left( \frac{\hat{V}_t(w)\eta^2 + 1}{\alpha^2}(1 + o(1)) \right)$$

$$= \left( \frac{\hat{V}_t(w)\eta^2 + 1}{\eta^2} + o(\hat{V}_t(w)) \right) \left( \log\left( \frac{\hat{V}_t(w)\eta^2 + 1}{\alpha^2} \right) + \log(1 + o(1)) \right).$$

Since $\log(1 + x) = x + o(1)$ for small $x$, we have $\log(1 + o(1)) = o(1)$. Expanding and noting that the cross terms are $o(\hat{V}_t(w) \log \hat{V}_t(w))$:

$$b(V_t(w); \alpha)^2 = b(\hat{V}_t(w); \alpha)^2 + o(\hat{V}_t(w) \log \hat{V}_t(w)).$$

This gives $b(V_t(w); \alpha)^2 / b(\hat{V}_t(w); \alpha)^2 = 1 + o(1)$, and since $(1 + o(1))^{1/2} = 1 + o(1)$:

$$b(V_t(w); \alpha) = b(\hat{V}_t(w); \alpha)(1 + o(1)).$$

Combined with the strong approximation error being $o_{a.s.}((V_t(w) \log \log V_t(w))^{1/2}) = o(b(V_t(w); \alpha))$, this establishes that $\hat{r}_t(w) \pm b(\hat{V}_t(w); \alpha)$ is a $(1 - \alpha)$ asymptotic confidence process for $r_t(w)$. $\qquad\square$

## A.3. Proof of Lemma 4.7

*Proof.* Since $(X_t)_t$ is a martingale with $X_0 = 0$, it follows that $\mathbb{E}[X_t] = \mathbb{E}[X_0] = 0$ for all $t \geq 0$. Consequently, the covariance reduces to the expectation of the product:

$$\text{Cov}(X_s, X_t) = \mathbb{E}[X_s X_t] - \mathbb{E}[X_s]\mathbb{E}[X_t] = \mathbb{E}[X_s X_t]. \tag{41}$$

By the Law of Iterated Expectations, we condition on $\mathcal{F}_s$:

$$\mathbb{E}[X_s X_t] = \mathbb{E}\left[\mathbb{E}[X_s X_t \mid \mathcal{F}_s]\right]. \tag{42}$$

Since $X_s$ is $\mathcal{F}_s$-measurable, it factors out of the conditional expectation:

$$\mathbb{E}\left[\mathbb{E}[X_s X_t \mid \mathcal{F}_s]\right] = \mathbb{E}\left[X_s \mathbb{E}[X_t \mid \mathcal{F}_s]\right]. \tag{43}$$

Applying the martingale property $\mathbb{E}[X_t \mid \mathcal{F}_s] = X_s$ for $s \leq t$, we obtain:

$$\mathbb{E}\left[X_s \mathbb{E}[X_t \mid \mathcal{F}_s]\right] = \mathbb{E}\left[X_s^2\right]. \tag{44}$$

Finally, noting that $\text{Var}(X_s) = \mathbb{E}[X_s^2] - (\mathbb{E}[X_s])^2 = \mathbb{E}[X_s^2]$, we conclude that $\text{Cov}(X_s, X_t) = \text{Var}(X_s)$. $\qquad\square$

## A.4. Proof of Covariance Formula (23)

*Proof.* Since units are randomized independently, $\text{Cov}(\hat{\Delta}_s, \hat{\Delta}_t) = \sum_{i \in \mathcal{E}_t} \text{Cov}(\hat{\Delta}_{is}, \hat{\Delta}_{it})$. For a single unit $i$, we have

$$\text{Cov}(\hat{\Delta}_{is}, \hat{\Delta}_{it}) = \mathbb{E}[\hat{\Delta}_{is}\hat{\Delta}_{it}] - \mathbb{E}[\hat{\Delta}_{is}]\mathbb{E}[\hat{\Delta}_{it}] = \mathbb{E}[\hat{\Delta}_{is}\hat{\Delta}_{it}] - \Delta_{is}\Delta_{it}. \tag{45}$$

Write the AIPW estimator as $\hat{\Delta}_{it} = (m_{it}(1) - m_{it}(0)) + A_{it}$, where

$$A_{it} = \frac{\mathbf{1}[w_i = 1]}{\pi_i(1)} e_{it}(1) - \frac{\mathbf{1}[w_i = 0]}{\pi_i(0)} e_{it}(0). \tag{46}$$

Note that $\mathbb{E}[A_{it}] = e_{it}(1) - e_{it}(0)$.

Expanding the product and taking expectations:

$$\begin{aligned}
\mathbb{E}[\hat{\Delta}_{is}\hat{\Delta}_{it}] = {}& (m_{is}(1) - m_{is}(0))(m_{it}(1) - m_{it}(0)) \\
& + (m_{is}(1) - m_{is}(0))(e_{it}(1) - e_{it}(0)) \\
& + (m_{it}(1) - m_{it}(0))(e_{is}(1) - e_{is}(0)) + \mathbb{E}[A_{is}A_{it}].
\end{aligned} \tag{47}$$

Similarly, since $\Delta_{it} = (m_{it}(1) - m_{it}(0)) + (e_{it}(1) - e_{it}(0))$:

$$\begin{aligned}
\Delta_{is}\Delta_{it} = {}& (m_{is}(1) - m_{is}(0))(m_{it}(1) - m_{it}(0)) \\
& + (m_{is}(1) - m_{is}(0))(e_{it}(1) - e_{it}(0)) \\
& + (m_{it}(1) - m_{it}(0))(e_{is}(1) - e_{is}(0)) \\
& + (e_{is}(1) - e_{is}(0))(e_{it}(1) - e_{it}(0)).
\end{aligned} \tag{48}$$

Therefore:

$$\mathbb{E}[\hat{\Delta}_{is}\hat{\Delta}_{it}] - \Delta_{is}\Delta_{it} = \mathbb{E}[A_{is}A_{it}] - (e_{is}(1) - e_{is}(0))(e_{it}(1) - e_{it}(0)). \tag{49}$$

To compute $\mathbb{E}[A_{is}A_{it}]$, note that when $w_i = 1$:

$$A_{is}A_{it} = \frac{e_{is}(1)e_{it}(1)}{\pi_i(1)^2}, \tag{50}$$

and when $w_i = 0$:

$$A_{is}A_{it} = \frac{e_{is}(0)e_{it}(0)}{\pi_i(0)^2}. \tag{51}$$

Thus:

$$\mathbb{E}[A_{is}A_{it}] = \pi_i(1) \cdot \frac{e_{is}(1)e_{it}(1)}{\pi_i(1)^2} + \pi_i(0) \cdot \frac{e_{is}(0)e_{it}(0)}{\pi_i(0)^2} = \frac{e_{is}(1)e_{it}(1)}{\pi_i(1)} + \frac{e_{is}(0)e_{it}(0)}{\pi_i(0)}. \tag{52}$$

Combining and summing over all units yields (23). □

### A.5. Proof of Proposition 4.12

*Proof.* Let $S = V_0 + V_1$ and $\rho_S = V_0/S$, so that $\rho_S \to \rho$ and $V_1/S \to 1 - \rho$. We first record the scaling of the denominator. Since

$$\frac{V_1}{1 - \pi} + \frac{V_0}{\pi} = S \left\{ \frac{1 - \rho_S}{1 - \pi} + \frac{\rho_S}{\pi} \right\}, \tag{53}$$

and the term in braces converges to

$$c_\rho := \frac{1 - \rho}{1 - \pi} + \frac{\rho}{\pi} > 0, \tag{54}$$

the normal-mixture boundary satisfies

$$\frac{b\left( \frac{V_1}{1-\pi} + \frac{V_0}{\pi}; \alpha \right)}{\sqrt{S \log S}} \to \sqrt{c_\rho}. \tag{55}$$

Indeed, this follows directly from $b(V; \delta) = \sqrt{V \log(V\eta^2/\delta^2)}(1 + o(1))$ whenever $V \to \infty$.

For the numerator, the same expansion gives

$$\frac{b(V_0; \alpha/2)}{\sqrt{S \log S}} \to \sqrt{\rho},$$
$$\frac{b(V_1; \alpha/2)}{\sqrt{S \log S}} \to \sqrt{1 - \rho}. \tag{56}$$

The endpoint cases are included in these limits: if, for example, $V_0 = o(S)$, then $V_0 \leq S$ and $b(V_0; \alpha/2)^2 = O((V_0 + 1)\log S) = o(S \log S)$. Therefore

$$\frac{b(V_0; \alpha/2) + b(V_1; \alpha/2)}{\sqrt{S \log S}} \to \sqrt{\rho} + \sqrt{1 - \rho}. \tag{57}$$

Dividing the numerator limit by the denominator limit gives

$$R_\pi(V_0, V_1) \to \frac{\sqrt{\rho} + \sqrt{1 - \rho}}{\sqrt{\rho/\pi + (1 - \rho)/(1 - \pi)}}. \tag{58}$$

Finally, the limiting ratio is at most one by Cauchy's inequality:

$$\left( \sqrt{\rho} + \sqrt{1 - \rho} \right)^2 \leq \left( \frac{\rho}{\pi} + \frac{1 - \rho}{1 - \pi} \right)(\pi + 1 - \pi). \tag{59}$$

The symmetric and asymmetric limits follow by substituting $\rho = 1/2$, $\rho = 0$, and $\rho = 1$. □

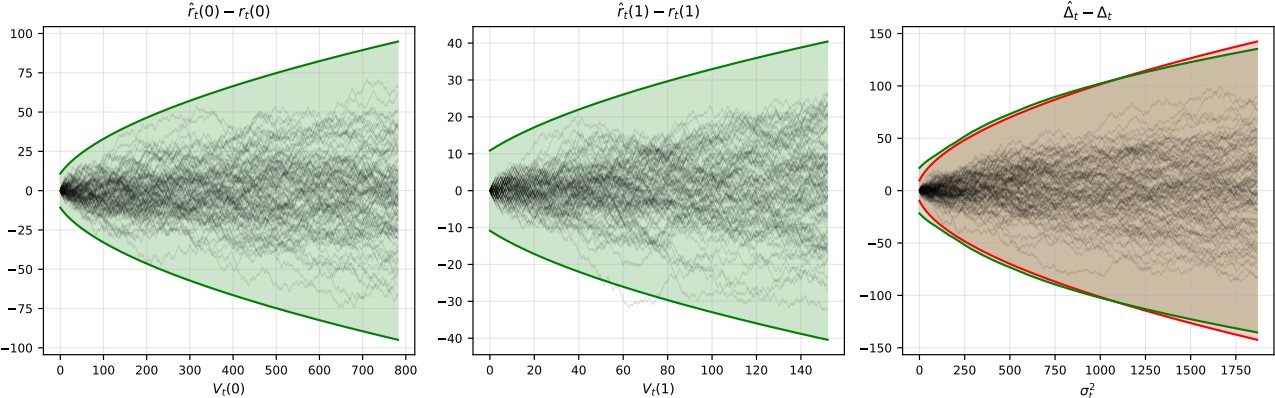

*Figure 2.* Sample paths of IPW estimation errors from 100 Monte Carlo replications of Dataset 5.1. Left panel: $\hat{r}_t(0) - r_t(0)$ plotted against the oracle variance clock $V_t(0)$, with boundary $\pm b(V_t(0); \alpha/2)$. Center panel: $\hat{r}_t(1) - r_t(1)$ plotted against $V_t(1)$, with boundary $\pm b(V_t(1); \alpha/2)$. Right panel: $\hat{\Delta}_t - \Delta_t$ plotted against the variance-upper-bound clock $\sigma_t^2$, with the union-bound boundary $\pm\{b(V_t(0); \alpha/2) + b(V_t(1); \alpha/2)\}$ in green and the benchmark boundary $\pm b(\sigma_t^2; \alpha)$ in red.

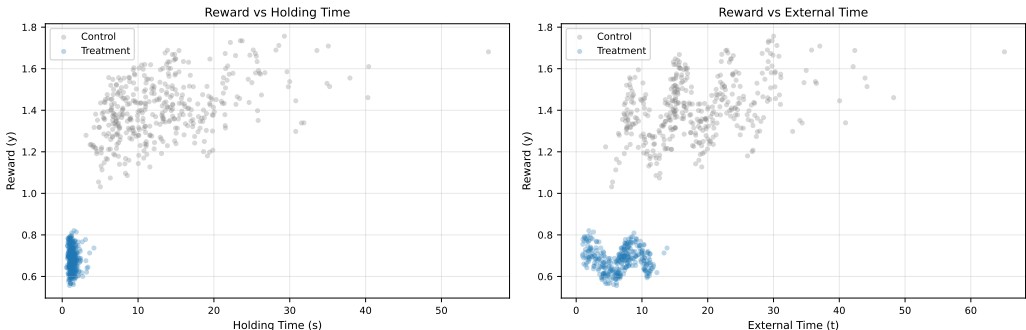

*Figure 3.* Illustration of outcome dependence on treatment, internal time, and external time for Dataset 5.1. The potential outcome $y_i(w)$ depends on treatment $w$ (which determines the baseline $\beta_w$), internal time $s_i = t_i(w) - E_i$ (time since entry, contributing a logarithmic growth), and external time $t_i(w)$ (contributing cyclical variation).

## B. Supplemental Figures

## C. Case Study: Out-of-Memory Errors

In software delivery, A/B "Canary" Tests are often used as quality control gates to compare the release candidate (treatment) against the existing software version (control) in a production environment (Lindon et al., 2022). Devices are randomized into a treatment arm at app-startup (staggered entry) and emit logging events whenever an error occurs.

Out-of-memory (OOM) errors are a particularly important class of software defects where delayed outcomes arise naturally. A memory leak occurs when a program allocates memory but fails to release it when no longer needed; over time, memory usage grows until the system cannot satisfy an allocation request, triggering an OOM error. The delay between when a device starts running the buggy software and when the OOM error manifests depends on usage patterns and available memory - some devices may crash within minutes, others after hours. This makes OOM errors an ideal application for our methodology: the outcome (an OOM crash) is inherently delayed.

Anytime-valid inference is imperative for this setting. Due to the large-scale nature of such technology experiments, it is crucial to abort the canary test and roll back the release if the treatment arm is significantly worse than the control arm. A number of anytime-valid rank-based statistics could be considered for this setting (ter Schure et al., 2024; Lindon & Malek, 2022; Gu & Lai, 1991); however, these all assume a proportional hazards model for the hazard rate of errors, which is

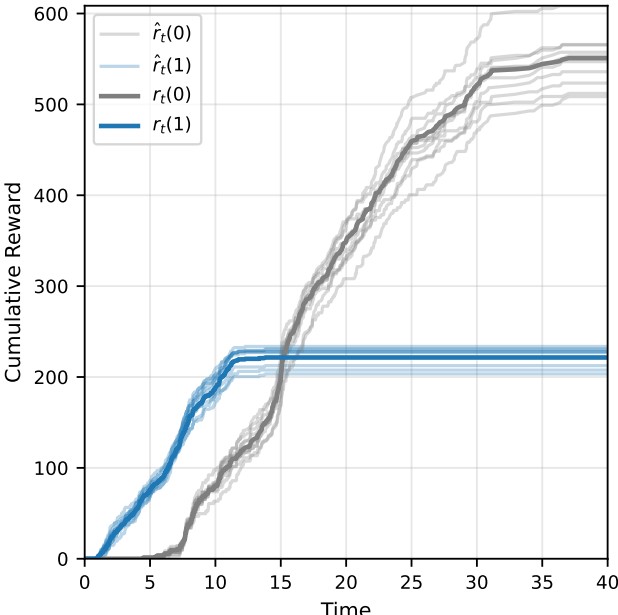

*Figure 4.* Ten realizations of $\hat{r}_t(1)$ and $\hat{r}_t(0)$ obtained by resampling treatment indicators $w_i$, compared with true values $r_t(1)$ and $r_t(0)$, for Dataset 5.1.

violated when resource depletion drives event timing.

We apply our methodology here in the special case $y_i(w) = 1$ for all $i$ and $w$, yielding a simple count-based approach. This addresses the counterfactual question "How many unique devices would have encountered an OOM error by the current time $t$, had they all been assigned to the treatment arm?". This corresponds to a counterfactual reality in which the release engineer had simply deployed the release candidate to all global production devices. The same question can be asked for the control arm, and compared with the treatment.

We apply this methodology to telemetry data from a real production canary test, where devices were randomized 50/50 between the release candidate and the existing software version. Using the IPW estimator, Figure 6 shows the design-based confidence sequences constructed using Theorems 4.6 and 4.9 and the sequential $p$-value from Corollary 4.10. Almost immediately, we can detect an increase in OOM errors among devices running the newer treatment software.

### C.1. Comparison to Lindon & Kallus (2025)

Lindon & Kallus (2025) model the arrival of OOM errors through time as an inhomogeneous Poisson process with cumulative intensity $\Lambda_t(w)$, constructing confidence sequences via conjugate Gamma mixtures. In the restrictive special case when assignment probabilities are *constant* ($\pi_i(w) = \pi(w)$ for all $i$), the design-based IPW estimator and the Poisson-based estimator are closely related.

Under the design-based framework, the IPW estimator for arm $w$ is

$$\hat{r}_t(w) = \sum_{i:w_i=w,t_i\leq t} \frac{y_i}{\pi_i(w)} = \frac{N_t(w)}{\pi(w)}, \tag{60}$$

where $N_t(w)$ is the observed count in arm $w$ by time $t$. Under the Poisson model, $N_t(w) \sim \text{Poisson}(\Lambda_t(w))$ is treated as the realization of a time-inhomogeneous Poisson process with cumulative intensity $\Lambda_t(w) = \mathbb{E}[N_t(w)]$. Lindon & Kallus (2025) construct confidence processes on $\Lambda_t(1)$, $\Lambda_t(0)$, and the difference $\Lambda_t(1) - \Lambda_t(0)$, which can be simply rescaled by $1/\pi(w)$ to adjust for unbalanced assignment. Figure 7 shows the Poisson-based confidence sequences on the same data, scaled by $1/\pi(w) = 2$.

Both approaches yield identical point estimates when the Poisson estimates are scaled by $1/\pi(w)$. They differ, however,

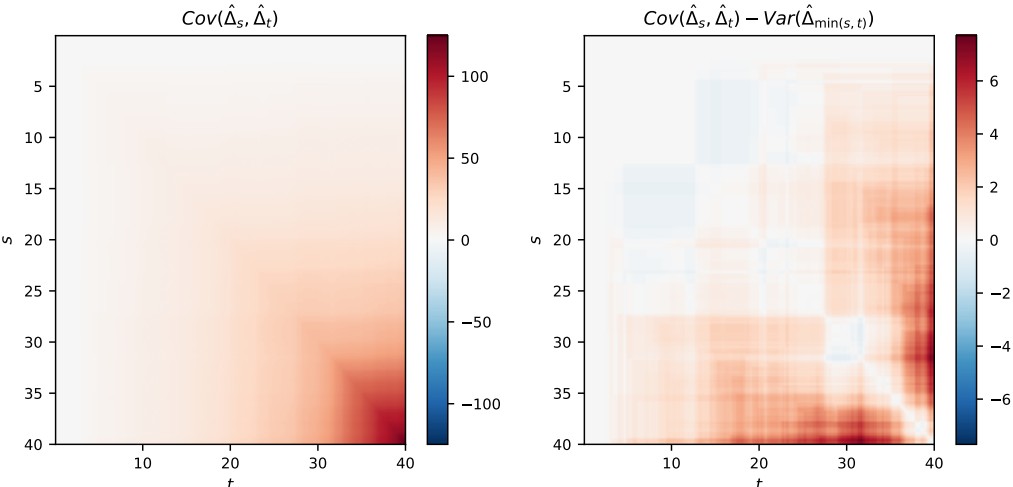

*Figure 5.* Example of martingale violation for $\mathrm{Cov}(\hat{\Delta}_s, \hat{\Delta}_t) \neq \mathrm{Var}(\hat{\Delta}_s)$ for Dataset 5.1, using random augmentation terms $m_{it}(0) \overset{iid}{\sim}$ Uniform$(0.5, 2.0)$ and $m_{it}(1) \overset{iid}{\sim}$ Uniform$(0.2, 1.0)$.

fundamentally in how they handle uncertainty. The uncertainty in Lindon & Kallus (2025) is *model-based*, whereas the uncertainty in our approach is *design-based*, leveraging the randomization of treatment assignment. The former requires the modeling assumption of a *Poisson* arrival process, which is a critical assumption as it allows a martingale property to be established via the memoryless independent increments property. However, the Poisson model is not always appropriate, and, as we demonstrate here, not necessary. We achieve the martingale property in the design-based framework via randomization and SUTVA; no modeling assumptions are required. Moreover, the Poisson approach does not naturally accommodate varying assignment probabilities $\pi_i(w)$ across units, which precludes release engineers from using adaptive experimental designs to carefully roll-out new software. In short, the design-based approach makes no distributional assumptions and is more flexible.

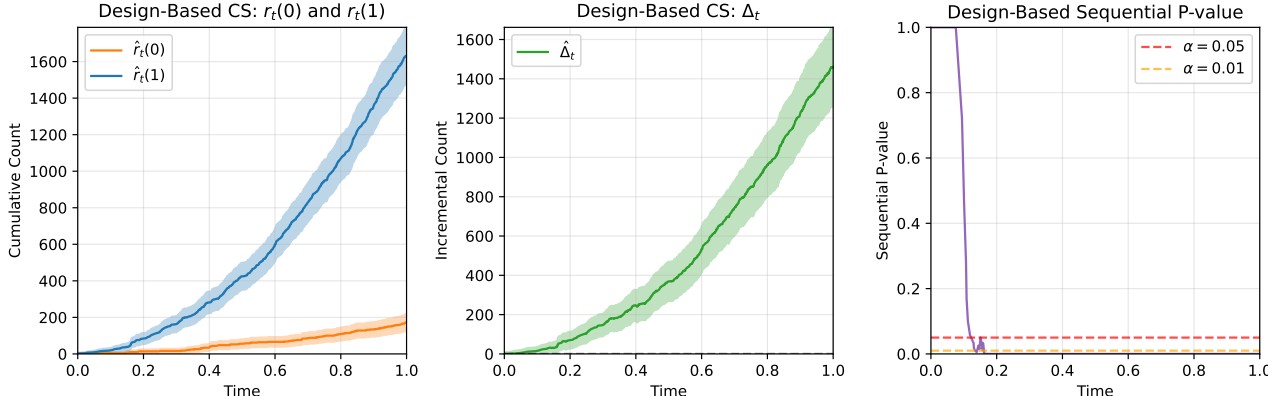

*Figure 6.* Design-based confidence sequences for OOM error counts. Left: single-arm confidence sequences for $r_t(0)$ and $r_t(1)$. Center: treatment effect confidence sequence for $\Delta_t = r_t(1) - r_t(0)$. Right: sequential p-value for testing $H_0 : \Delta_t = 0$.

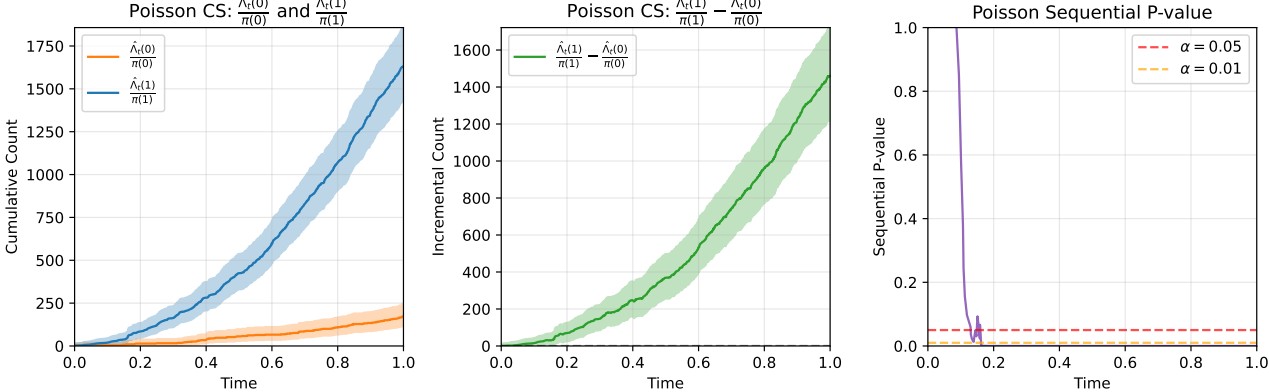

*Figure 7.* Poisson process confidence sequences for the same data, scaled by $1/\pi(w) = 2$ to target the same estimand $r_t(w)$ as the design-based approach. Left: confidence sequences for $r_t(0)$ and $r_t(1)$. Center: confidence sequence for $\Delta_t = r_t(1) - r_t(0)$. Right: sequential p-value. The point estimates are identical to Figure 6; the confidence intervals differ only due to the distinct boundary functions used by each method.

