# OpenReview forum: "Design-Based Anytime-Valid Inference for Randomized Experiments with Delayed Outcomes and Staggered Entry"
_ICML.cc/2026/Conference — ICML 2026 regular_

### Official Review · Reviewer_xGAe · 2026-03-13

**Soundness:** 3
**Presentation:** 3
**Significance:** 3
**Originality:** 3
**Overall Recommendation:** 4
**Confidence:** 3

**Summary:**

The paper develops a design-based framework for anytime-valid inference in randomized experiments with delayed outcomes, where treatment can affect both the timing and the magnitude of outcomes. The authors target a calendar-time causal estimand, namely the sample cumulative reward process, and show that a single-arm IPW estimator admits a martingale structure under a carefully chosen single-arm filtration, enabling asymptotic confidence sequences, while no filtration renders the treatment-effect error a martingale. They address the latter via a principled union bound across arms. The synthetic simulation provides empirical evidence of time-uniform coverage for the IPW construction under the paper's nonstationary, staggered-entry setup, and the telemetry case study illustrates practical use. The theory also analyzes an asymptotic regime in which the union-bound construction can be tighter than a variance-upper-bound-based alternative.

**Compliance With Llm Reviewing Policy:**

Affirmed.

**Final Justification:**

My questions have been answered. I will keep my positive overall recommendation.

**Key Questions For Authors:**

1. How should readers interpret the finite-sample reliability of the asymptotic confidence sequences, and do you have additional finite-sample calibration evidence beyond the current simulation?
2. Can you provide additional empirical analysis across different delay regimes?

I am not fully familiar with this line of work, and I would be open to updating my assessment after reading the authors' clarification.

**Limitations:**

The paper already makes the asymptotic nature explicit, but it should discuss the practical implications of asymptotic rather than finite-sample validity, together with the one-delayed-event-per-unit restriction and the limited empirical scope.
The paper could also discuss the practical implications of the filtration construction more explicitly.

**Strengths And Weaknesses:**

Strengths

- The core design-based argument appears sound, including the single-arm martingale construction, the impossibility result, and the union-bound resolution.
- The paper makes a genuine conceptual contribution around anytime-valid inference with delayed outcomes.
- The problem is important for online experimentation.
- The paper explicitly models outcome timing as a potential outcome and accommodates treatment effects on both arrival times and values under nonstationarity and staggered entry.
- The filtration analysis is particularly insightful, especially the role of the union bound across arms.
- The synthetic study shows that pointwise intervals can fail and that the proposed IPW procedure shows high empirical time-uniform coverage, while the telemetry case study is a useful real-world illustration.
- The exposition of the designer, observer, and single-arm filtrations is clear and well motivated.

Weaknesses

- The guarantees are asymptotic, so finite-sample behavior is supported mainly by the simulation study.
- The empirical evaluation is useful, but it could still be broader in scope.
- The scope is limited by the one-delayed-event-per-unit setup.
- The paper could provide more practical guidance on choosing eta in applied settings.

---

> ### Author Rebuttal · Authors · 2026-03-28
>
> We sincerely thank the reviewer for their careful reading and for recognizing the value of our core contributions.
>
> **W1/W4/Q1: Finite Sample Reliability**
> This is an excellent question. There is a tradeoff between finite sample reliability and conservatism. There are indeed nonparametric confidence sequences with finite-sample guarantees (see the Empirical Bernstein confidence sequence of Howard et al. 2021). However, the price paid for the finite-sample guarantee is a wider (less precise) confidence sequence. This has been documented in other works such as Bibaut et al. 2022, who note that nonparametric finite-sample concentration bounds *"aren't as commonly used in applied statistical practice"* because they are *"over-conservative in general"*.
>
> Similarly Waudby-Smith et al. 2024 writes:
>
> *Moreover, notice that nonasymptotic CSs appear to be conservative, while our AsympCSs are much tighter and have miscoverage rates approaching α (as expected in light of Theorem 2.8).*
>
> Both the aforementioned papers study the tightness of the $1-\alpha$ coverage guarantee across varying asymptotic regimes (see Theorem 2.8 of Waudby-Smith et al. and Theorem 1 of Bibaut et al.) and provide compelling empirical results.
>
> Asymptotic confidence sequences via strong approximations have been seeing steady adoption across the literature. Noted works are:
> - Ham et al. 2022,
> - Maharaj et al 2023 "Anytime-Valid Confidence Sequences in an Enterprise A/B Testing Platform",
>
> demonstrating compelling empirical 1-\alpha coverage calibration
>
> To provide some intuition, the confidence sequence must not mis-cover at early times $t$ before the asymptotics have "kicked in". This typically happens when $\alpha$ is small or when $\eta$ is small. Please see the next section:
>
> **W4 cont...: practical guidance on choosing eta in applied settings.**
> The parameter $\eta$ can be interpreted as controlling the rate at which the $\alpha$-budget is spent. A small value of $\eta$ trades a wider confidence sequence at small $t$ for a narrower confidence sequence at large $t$. In other words, a small $\eta$ spends little $\alpha$ in the early parts of the experiment at small $t$, resulting in an initially-wider (and subsequently tighter) confidence sequence, providing robust protection against early mis-coverings as we wait for the asymptotics to kick in. Moreover, in large internet companies performing A/B tests, $\eta$ is already chosen *very* small to optimize it for the kinds of effect sizes observed in online experimentation. In these applications, the difference between treatment and control are often extremely tiny, so it makes no sense to spend $\alpha$ early at small sample sizes - it is much better to retain it and spend it at larger sample sizes. This strategy optimizes for detecting small effects, and provides amply time for asymptotic arguments to be justified.
> We would be happy to add a discussion on choosing $\eta$ and cite prior discussions. For example, Waudby-Smith et al. 2024 has a dedicated Appendix Section B.2 on choosing the hyperparameter $\eta$ in accordance with the practitioners expectaiton of the effect size.
>
>
> **W2/Q2: The empirical evaluation is useful, but it could still be broader in scope.**
> We appreciate this feedback. Our intention with the current synthetic simulation was to provide an adversarial, worst-case scenario (extreme time-delay + complex nonstationary shocks ) to clearly differentiate our setting from prior art. To broaden the empirical evaluation, we will add a semi-synthetic experiment to the appendix leveraging our real-data OOM timings. Specifically, we will impute the unobserved potential outcomes: for each unit observed in control (treatment), we will sample $t_i(1)$ ($t_i(0)$) with replacement (plus jitter) from the treatment (control) ECDF. Once drawn, these potential outcomes act as the fixed ground truth for each unit, allowing us to compute the exact, true time-varying treatment effect $\Delta_t$. We will then repeatedly resample treatment assignments and compute the empirical time-uniform coverage probability via Monte Carlo. This will directly demonstrate our confidence sequence's calibration under a highly realistic, organic delay regime.
>
> **W3: The scope is limited by the one-delayed-event-per-unit setup.**
> Our motivating applications indeed only had one event per unit, such as a conversion or purchase. While this is a wide class of experiment outcomes, nothing in our methodology precludes a unit from having multiple outcomes, provided each unit is independently *re-randomized*. This includes switch-back and timeseries experiments on the same unit, where a unit is repeatedly re-randomized into treatment and control. This would look very much like a new unit entering the experiment each time.
>
> **Significantly Strengthened Theorem 4.4 to Support AIPW Estimator** Please see our note "Q3" to reviewer DzfE where we provide a subtle correction that fully enables the AIPW estimator to be used.

---

> > ### Author Rebuttal · Reviewer_xGAe · 2026-04-01
> >
> > Thank you for the thoughtful rebuttal. The clarification on the asymptotic vs. finite-sample tradeoff and the practical role of $\eta$ was helpful, and I appreciate the discussion of possible broader empirical evaluation. I am keeping my overall positive recommendation.

---

### Official Review · Reviewer_FEr6 · 2026-03-13

**Soundness:** 2
**Presentation:** 2
**Significance:** 3
**Originality:** 4
**Overall Recommendation:** 4
**Confidence:** 2

**Summary:**

The paper considers the design of confidence sequences for a class of potential outcomes and treatment effect-like estimands.
Crucially, the setting involves outcome delay, where the outcome is only observed at a later, random, point in time.
This means that there is effectively a form of right-censorship happening at all times.
The authors tackle the problem by the means of the asymptotic confidence sequence framework of [Waudby-Smith et al., 2024] with a carefully constructed filtration; to this end, they base their asymptotics on an IPW-type estimator (and interestingly note that an AIPW-type estimator is unsuitable).

**Compliance With Llm Reviewing Policy:**

Affirmed.

**Final Justification:**

My view of the paper is tentatively more to the positive side, since the rebuttal did resolve my main concerns. However, it also became clear that I did not understand some parts of the original submission, and I was unable to reassess the whole paper post-rebuttal in light of this, so the confidence in my assessment is rather low.

**Key Questions For Authors:**

Could the authors please clarify what is going on with the estimand and the formal definition of the confidence sequence being used? Some clarification on the proof of e.g. Theorem 4.6 would also be appreciated.

**Limitations:**

All relevant limitations were appropriately discussed.

**Strengths And Weaknesses:**

The paper tackles the very important problem of doing sequential anytime-valid inference when there is outcome delay.
Many real-world settings fall under this regime, yet nevertheless it still hasn't been well studied.
This paper makes significant strides in this direction.

That said, I have a couple of concerns/questions:

- I am a bit uncomfortable with the target estimand. As far as I can tell, the 'estimand' $r_t(w)$ (as well as $\Delta_t = r_t(1) - r_t(0)$) are random quantities. Yet in e.g. Theorem 4.6 the authors claim to construct an "asymptotic confidence process for $r_t(w)$". Could the authors clarify what exactly they mean by this? I was unable to discern this from my quick-ish look at the proof of the theorem, which makes me a bit uneasy with regards to its correctness (but hopefully the authors could help elucidate). My current best guess is that this is indeed a random estimand and that when appealing to Ville's inequality this randomness is included by the means of the particular martingale that was constructed.
- Relatedly, in line 135/136 it is said "notice our estimand is not a property of a distribution, [...]"; but formally speaking, it can be written as a property of the true data distribution, no? In the same sense that a mean is a functional of the distribution. Though I imagine the authors refer to the fact that this is not simply a parameter of the distribution? Or is it something else?
- It is quite curious that an IPW-type estimator is required. Does this imply that the proposed method is perhaps suboptimal in some respect? (For the record, I think the proposed solution is still relevant and interesting regardless of any potential suboptimality.)
- In Corollary 4.10 the authors propose to compute a p-value from their confidence sequences. Would it not be better to construct an e-value? (I imagine this could be done with relative ease.) The construction of an e-value here would still give a p-value via its reciprocal, while also enabling the use of the stronger e-value-based procedures e.g. for FDR control, as well as allowing for the significance level to be set a posteriori.
- Additionally, I think that the presentation could be considerably improved. For example, I think it would be great if before starting to talk about martingales there was some motivation for (i) why look at martingales, and (ii) why look at *that* martingale in particular, rather than e.g. some e-process directly.

Overall, I'm giving a borderline reject score mainly due to my discomfort with the estimands and the correctness of the main theorem(s). Should the authors resolve this, I'd be happy to raise my score.

---

> ### Author Rebuttal · Authors · 2026-03-25
>
> We thank the reviewer for their careful reading. We are glad to address the concern about the estimand, which we believe stems from a natural but resolvable confusion between the design-based and superpopulation perspectives.
>
> **Estimand and correctness of theorems.** In the potential outcomes framework (Neyman 1923; Rubin 1974; Imbens & Rubin 2015), potential outcomes are *fixed* attributes of each unit — this is standard in both the design-based and superpopulation settings. Since $y_i(w)$, $t_i(w)$, and $E_i$ are all fixed, the estimand $r_t(w) = \sum_{i \in \mathcal{E}_t} y_i(w) \mathbf{1}[t_i(w) \leq t]$ is deterministic. Treatment assignment $W_i$ is the sole source of randomness, and it does not appear in $r_t(w)$. The *estimator* $\hat{r}_t(w)$ is random (through $W_i$), but the *estimand* is not. The confidence sequence therefore has the standard interpretation: a random interval covering a fixed target, uniformly over time. There is no additional subtlety in appealing to Ville's inequality — the process $M_t(w) = \hat{r}_t(w) - r_t(w)$ is a martingale, where the randomness comes entirely from treatment assignment. For a fuller discussion of the design-based vs. superpopulation distinction and why we adopt the former, please see our rebuttal item "W1" to Reviewer KFyR
>
> **"Not a property of a distribution."** We mean this literally: the probability space $\Omega = \lbrace 0,1\rbrace^\infty$, the set of possible treatment assignments, while the potential outcomes $t_i(w), y_i(w)$ are fixed, non-random constants.
> Our target estimand $r_t(w)$ is a deterministic function of the *sample* potential outcomes and contains no random variables.
> The *observed* outcome $r_{it} = w_i r_{it}(1) + (1-w_i)r_{it}(0)$ and estimator $\hat r_{it}(w)$ are random because they depend on the random assignment indicator $w_i$. Here we *condition* on the units in the experiment.
> In contrast, the superpopulation framework does not *condition* on the units, and treats them as a random sample from a superpopulation $P$. Although the potential outcomes for each unit are fixed, the units that enter are random through random sampling, hence the potential outcomes for the units in the experiment have two sources of uncertainty: assignment and sampling. In this framework, the *population* estimand is  $E_P[r_{it}(1) - r_{it}(0)]$, where $P$ is the superpopulation distribution.
> We do not target a *population* causal estimand, however, as a stationary target superpopulation is ill-defined in our setting.
> In online settings, nonstationarity (seasonality, shocks) and staggered entry (cohort heterogeneity) make such a distribution ill-defined. There is no well-defined future superpopulation to generalize the inference to.
> The design-based perspective sidesteps this by conditioning on the realized units and relying only on the physical act of randomization. In doing so it answers the question: "what is the treatment effect on the units in the experiment?" as opposed to  "what is the treatment effect on some hypothetical infinite superpopulation?".
>
> When A/B testing on members, one really cares about the treatment effect *on these members*, not individuals from a hypothetical superpopulation.
>
>
> **IPW required / suboptimality - AIPW is fully supported** We have significantly strengthened this result.
> The version you reviewed suggested only IPW preserved the martingale property; however, we have since corrected a subtle $\leq$ vs $<$ boundary error in the proof of Theorem 4.1. Please see our rebuttal item "Q3" to Reviewer DzfE for the corrected theorem statement and proof. Theorem 4.1 proves that event-time AIPW, where augmentation activates at $t_i(w)$ and remains constant thereafter, is a valid martingale. This is a significant update as it means our method is not suboptimal. It permits full variance reduction via covariate adjustment. This theoretical correction now aligns perfectly with the strong empirical performance of the AIPW estimator shown in our simulations.
>
> **E-values.** Thank you for this suggestion. Indeed, for a stopping time $\tau$, the random variable that is $1/p_\tau$ (i.e. the reciprocal of the stopped *sequential* p-value) is an $e$-value. It is a good idea to comment on the connection to $e$-processes and $e$-values to strengthen connections with the existing literature. We will make this connection explicit in the
>   revision.
>
> **Presentation / motivation for martingales.** As this point was brought up by another reviewer, we recognize we must provide better motivation. We agree and will add a discussion of why the martingale property is necessary (Ville's inequality for time-uniform bounds) and why we construct this particular martingale (via design-based randomization and the single-arm filtration). The key reference is Ramdas et al. (2020) "Admissible anytime-valid sequential inference must rely on nonnegative martingales". Please See also item "W2" of the rebuttal to reviewer KFyR.

---

> > ### Author Rebuttal · Reviewer_FEr6 · 2026-04-04
> >
> > I'd like to thank the authors for their response.
> > I understand the underlying modelling and the arguments a bit better now, and am raising my score accordingly.
> > The other changes are also appreciated.
> >
> > That said, it's apparent to me that my original understanding of the paper was rather off-the-mark, and I was unfortunately unable to fully reassess the work in light of the authors' clarifications in time for this response; I have also realized that I am perhaps unfamiliar with some key ideas being used in this work. Therefore, I am also lowering my confidence.
> > For a future revision, I suggest that the authors be *extremely* explicit about the randomness from the moment that the variables are introduced, and reinforce this idea throughout the text; this may help avoid confusions of this kind.

---

> > > ### Author Response · Authors · 2026-04-06
> > >
> > > Thank you for this feedback and for taking the time to revisit the paper in light of our rebuttal. We really appreciate your acknowledgement that the concerns were fully resolved and that you would raise your score accordingly. We noticed that the visible review still appears to show the earlier score; could you please update your score in the system before the rebuttal period ends on April 7? Thank you again for your time and constructive review. We will certainly follow the reviewer’s suggestion to reiterate and clarify the sources of randomness throughout the text.

---

### Official Review · Reviewer_DzfE · 2026-03-13

**Soundness:** 3
**Presentation:** 3
**Significance:** 3
**Originality:** 3
**Overall Recommendation:** 4
**Confidence:** 3

**Summary:**

This paper studies the problem of any-time valid inference in the presence of outcome delays, which is a very common problem in online experimentation. The authors adopted a design-based framework, where both the outcome timing and value are fixed potential outcomes, and randomness is introduced by treatment assignment only. Then the authors discussed the validity of different estimators and variance bounds.

**Compliance With Llm Reviewing Policy:**

Affirmed.

**Final Justification:**

My questions have been answered. I will keep my positive score.

**Key Questions For Authors:**

1. Can the framework handle infinite delays? meaning, some outcomes are censored and may never appear in the experiments.

2. One strange target here is that, by defining the $r_t$ variable, the estimator $\Delta_t$ is unbiased to comparison of the true $r_t$'s, instead of the original potential outcomes. In other words, the estimands are not the original causal effects anymore, but some scaled version that incorporates the delays. This is not an ideal formulation; I wonder if there is a way to show that the constructed intervals are also robust (maybe after mild modification) to contrasts of original potential outcomes?

3. For theorem 4.4, I understand that the AIPW estimator might fails to be a valid martingale on $\mathcal{F}_t(w)$, as some $m_{i,t}$ might still be well defined when outcomes are missing. I thought it would suffice to form a martingale if the $m_{i,t}$ is defined to be zero when delay occurs, but it seems that, based on the statement of the theorem, we would require all $m_{i,t} = 0$. What is the intuition behind that?

4. The current framework seems to focus on only one treatment assignment at the very beginning of the experiment; yet in reality, adaptive experiments are pretty common in a long-term horizon where multiple treatments are usually assigned along the way. Does the current framework cover that case as well? Or is it only applicable to a randomized trial?

5. I think the evaluation comparison right now is somewhat limited. The reason is that there are no other strong baselines beyond the naive point-wise CIs, which apparently work poorly. How about a baseline, for example, that approximates the continuous process with discretization and applies some alpha-spending strategies?

**Limitations:**

yes

**Strengths And Weaknesses:**

Strength:

1. The paper studies a very important problem in practice: how should people handle delay when monitoring an online platform? Delays are usually unpredictable, adversarial, sometimes informative, and how to ensure the system is robust to them or even takes advantage of them, is always an interesting question to explore.

2. The framework is novel and interesting. I have also read/written papers on adaptive randomized experiments with delays, but haven't explored such directions in a design-based perspective.

3. Interesting theoretical results: the authors proved some negative results on the very commonly used AIPW estimators, which is an interesting result as AIPW/IPW seem to be the default pair that works for most causal inference problems. Having a negative result inspires more thoughts on their difference.

Weakness: I want to point out several weaknesses in the next section (key questions).

---

> ### Author Rebuttal · Authors · 2026-03-25
>
> Thank you to the reviewer for their excellent feedback. The reviewer has touched upon an important point regarding augmentation, please see below:
>
> **Q3: Theorem 4.4: AIPW Estimator - Updated Result**
> The reviewed version stated that only IPW ($m_{it}(w) = 0$ for all $i, t$) preserves the martingale property. Shortly after submission, we identified
>    and corrected a subtle error in the "only if" direction of that proof: the boundary condition required $m_{it}(w) = 0$ for $t \leq t_i(w)$, when in fact only $t <
>    t_i(w)$ is necessary. The distinction is a single boundary point at the event time itself, $t = t_i(w)$, where the treatment assignment $w_i$ is freshly revealed, so $Z_i$ has zero conditional
>    mean and a nonzero augmentation at that instant is perfectly valid. Theorem 4.4 now provides the full characterization: the estimation error is a martingale if and only
>     if $m_{it}(w) = 0$ for $t < t_i(w)$ and $m_{it}(w)$ is constant on $[t_i(w), \infty)$. This $< $ versus $\leq$ correction opens up the entire class of event-time
>    augmentations $m_{it}(w) = f_i(w) \cdot \mathbf{1}[t_i(w) \leq t]$.
>
> The new theorem reads:
>
> **Theorem 4.4 (updated).**
>   The cumulative estimation error process $M_t(w)$ in equation 15 is a martingale with respect to $F_t(w)$ if and only if $m_{it}(w) = 0$ for all $t < t_i(w)$ and $m_{it}(w)$ is constant on $[t_i(w), \infty)$, for each $i$.
>
> We provide the proof of the if direction below (and are happy to provide the only if proof upon request):
> Assume $m_{it}(w) = 0$ for all $t < t_i(w)$ and $m_{it}(w)$ is constant on $[t_i(w), \infty)$. For $t < t_i(w)$, the residual is $e_{it}(w) = 0 - 0 =
>   0$, so $Z_i e_{it}(w) = 0$ regardless of the unobserved $w_i$ (adapted). For $t \geq t_i(w)$, $w_i$ is $F_t(w)$-measurable and $e_{it}(w) = y_i(w) -
>   m_{i,t_i(w)}(w)$ is a fixed constant (adapted). Since $m_{it}(w)$ is constant on $[t_i(w), \infty)$, the process jumps only at $t_i(w)$, with increment $Z_i \cdot (y_i(w) -
>    m_{i,t_i(w)}(w))$. Since $w_i$ is independent of $F_{t_i(w)-}(w)$, the conditional expectation of $Z_i$ is zero, and the fixed factor $(y_i(w) - m_{i,t_i(w)}(w))$
>   factors out. Hence we have zero-mean increments at all jump times and constant between jumps.
>
> **Q1: Can the framework handle infinite delays?** Yes — if $t_i(w) = \infty$, then $r_{it}(w) = y_i(w) \cdot \mathbf{1}[\infty \leq t] = 0$ for all finite $t$. The unit simply never contributes. The estimand remains well-defined, and procedurally this unit never enters (or simply contributes zero) to the estimator and variance.
>
> **Q2: "The estimands are not the original causal effects anymore."** By "original" causal effects we understand the eventual rewards $\lim_{t \to \infty} r_{it}(w) = y_i(w)$. But at any time $t < t_i(w)$, one cannot know $y_i(w)$ without modeling the unobserved future — and any such model requires strong assumptions that are violated by the nonstationarity in our setting. This paper explores what can be reported without making such assumptions. Our estimand targets what can honestly be inferred in real time: "how much cumulative revenue would we have generated by the current time $t$ had all units been assigned to treatment versus control?" This is directly interpretable to business stakeholders and avoids extrapolation. Note that classical fixed-horizon tests face the same constraint — they too can only compare revenue accumulated by the time of analysis, not eventual value.
>
> **Q4 (Adaptive experiments)**: The framework handles heterogeneous assignment probabilities $\pi_i(w)$ across units, which can be dynamically driven by adaptive experimental designs (e.g. for regret minimization or best arm identification) as new units enter the experiment. Alternatively if a previously assigned unit is re-randomized into a new arm, then as long as the randomization remains independent this would be no different to a new unit entering the experiment.
>
> **Q5 (Baselines)**: This is a common question that is heavily debated in the sequential analysis community, and both methods are appropriate in different circumstances. If a practitioner knows apriori the number of interim analyses they wish to perform, and at which times, then group-sequential testing methods are reasonable. However, they fail to support optional continuation: If one reaches the final interim analysis, and no strictly significant difference is observed, then the practitioner may be tempted to collect more data and "wait and see". Unfortunately, the alpha-spending budget had already spent the entire alpha, so this optional continuation is not permitted. Anytime-Valid methods, on the other hand, supports optional continuation. Please see section 8.1.2 of *Ramdas, Grünwald, Vovk & Shafer (2023), "Game-Theoretic Statistics and Safe Anytime-Valid Inference," Statistical Science* for further discussion.

---

> > ### Author Rebuttal · Reviewer_DzfE · 2026-04-01
> >
> > Thank you for the responses. I am happy with most of the statement. For Q4, I think I am referring to the setting where units are rerandomized based on their own history of treatments/covariates/outcomes, which is non-independent and definitely cannot be treated as an independent unit draw. So the current framework cannot cover this scenario, right? To be clear, I am totally fine with this; I just want to understand the limitations.

---

> > > ### Author Response · Authors · 2026-04-01
> > >
> > > Thank you for the clarification. After some additional thought on this extension, we totally agree with the reviewer that the scope of the paper is limited to inter-unit adaptation (where the probability $\pi_i$ for a new unit depends on the history of previous units), and does not currently cover intra-unit rerandomization.
> > >
> > > When a single unit is rerandomized multiple times, the potential outcomes now depend on the entire assignment sequence $w_i = (w_{i1}, w_{i2}, w_{i3}, \dots)$. This acknowledges that past assignments can affect influence current outcomes. In such settings, the causal estimand is less obvious than comparing treatment to control, as we must compare one assignment sequence $w_i$ to another sequence $w_i'$. In the design-based time series literature (Bojinov & Shephard, 2019 [1]), the authors consider the Contemporaneous Causal Effect (CTE), defined as $Y_i(w_{i,1:t-1},1) - Y_i(w_{i,1:t-1},0)$ for unit i.
> > >
> > > Extending our 'delayed outcome' model to this setting would require defining the outcome timing $t_i$ as an assignment-path-dependent potential outcome. This would require a significant reframing which our current results do not address. We will clarify in the final version that our martingale results are currently optimized for single-assignment units and Best-Arm Identification/MAB settings where timing is tied to a primary intervention.
> > >
> > > [1] Bojinov, I. and Shephard, N. (2019). "Time Series Experiments and Causal Estimands: Exact Randomization Tests and Trading." Journal of the American Statistical Association.

---

### Official Review · Reviewer_KFyR · 2026-03-15

**Soundness:** 3
**Presentation:** 2
**Significance:** 3
**Originality:** 3
**Overall Recommendation:** 4
**Confidence:** 2

**Summary:**

The paper studies the problem of average treatment effect estimation in temporal settings, where we care about the value of the outcome (here, reward) and the time the outcome is observed, which might be affected by the treatment, i.e., $(Y(1), t(1), Y(0), t(0))$.
The authors want to construct an estimator for ATE that is valid at any given time during the experiment.
They assume that, given the treatment assignment, the reward and time are deterministic, so the only randomness in the experiment stems from the treatment assignment.
Their goal is to construct a confidence sequence for the expected reward that is valid at any given time.

They show that the expected reward on the treated, at any given time $t$, if estimated by the IPW estimator, is a martingale, under an appropriate filtration while, if we use the AIPW estimator, there is no filtration for which it can be a martingale.
Thus, using IPW, we can get the desired confidence sequence.
However, they show that there is no filtration that can make the difference between the expected reward on the treated and untreated (equivalent of ATE) a martingale. Thus, they employ the estimators on the treated and untreated separately, and get the union bound of their confidence sequences as the confidence sequence of the final, treatment effect estimator.

Finally, they support their findings with experimental results that show that, although AIPW is not a martingale under any filtration, it exhibits good confidence sequence in practice, indicating that it behaves approximately like a martingale.

**Compliance With Llm Reviewing Policy:**

Affirmed.

**Final Justification:**

I beleive the paper introduces an important model for temporal causal inference and does a good job to propose strategies to address it. The comments after the rebuttal convinced of the motivation of the problem. However, I think that the techniques used are presented in a non-intutive way and the authors could make a better job presenting their approach and its limitations, as they committed to do in the rebuttal. I should say I am not an expert in the field, so maybe their techniques are standard and the presentation they have in the paper suffices.

**Key Questions For Authors:**

1. Could you provide examples when you define the three proposed filtrations? I believe this could add to the readability of the paper and provide good intuition.

2. I believe the notation used should be formally defined in the paper (or at least in the appendix). For example, in lines 234-238, the authors refer to prior works for the parameters $\theta, L$, but I think this is confusing and incomplete.

**Limitations:**

yes

**Strengths And Weaknesses:**

Strengths:
1. The lack of a filtration that makes ATE a martingale, forces the authors to use a union bound argument to control the confidence sequence of their estimator. The authors make the connection between this phenomenon and a similar one in the standard, sequential setting, where we need to bound the variance of the ATE estimator in a worst case, because we cannot control/estimate the dependence of the cross terms. The authors make the connection between the two cases (Remark 4.11) which is an interesting phenomenon.
2. Moreover, the authors observe that in the case hey study, the use of the union bound, although a worst-case approach, gives tighter upper bound when there is asymmetry in the variance clocks for each treatment value.


Weaknesses:
1. The authors model their problem assuming there is no randomness in the outcome or the time it is observed, given the value of the treatment. This assumption seems a bit restrictive. I would like to understand how it compares to the standard literature of (non-temporal) ATE estimation as well as why it is ok to make this assumption from a practical point of view.
2. The paper should have a better explanation of the techniques and assumptions. Throughout, the authors aim to show their estimators are martingales but they don't explain if this property is necessary and why. Later, when the ATE estimator is proven to not be a martingale, they can circumvent the problem and get a result by union bound. Why would this be a bad strategy in the case of AIPW, for example? Even if these arguments are standard in the literature I believe an overview/explanation should be given in the paper.

---

> ### Author Rebuttal · Authors · 2026-03-24
>
> We thank the reviewer for engaging with our work. We address each point below, which we believe will resolve the concerns raised.
>
>
> **W1: "The fixed potential outcomes assumption seems restrictive."** This is the standard assumption in the Neyman–Rubin potential outcomes framework (Fisher 1935; Neyman 1923; Imbens & Rubin 2015). In both the design-based and superpopulation causal inference frameworks, a given unit's potential outcomes $(y_i(0), y_i(1))$ are always fixed attributes of that unit. In the superpopulation framework, they appear random only because the identity of unit $i$ is a random draw from a superpopulation — one targets superpopulation causal estimands (e.g., the population average treatment effect) and uncertainty reflects both sampling and randomization. The design-based framework conditions on the realized sample and targets sample causal estimands (e.g., the sample average treatment effect) instead, removing the component of uncertainty from sampling. Hence, randomization is the only remaining source of uncertainty. All this amounts to is performing inference on a sample estimand instead of a superpopulation estimand. In doing so, we ask the question: "what was the effect of treatment on the units in the experiment?" (as opposed to the effect of treatment in some hypothetical superpopulation).
>
> We chose the design-based framework because the "superpopulation" is ill-defined in our context. A superpopulation approach would require specifying a stable joint distribution over outcome times and values $(t_i(w), y_i(w))$ — a distribution that is difficult to justify under the nonstationarity and staggered entry of our setting, where seasonality, promotions, and heterogeneous entry cohorts make the composition of the experimental population time-varying. The design-based perspective sidesteps these issues entirely: validity requires only SUTVA and known assignment probabilities, both of which are guaranteed by the experimenter's design.
>
> Similarly, when A/B testing on subscribers/members, one really cares about the treatment effect on these individuals, not individuals from a hypothetical superpopulation.
>
> **W2: "The paper should have a better explanation of the techniques and assumptions."** We appreciate this feedback. Providing a full onboarding to anytime-valid inference techniques while presenting our own contributions within the 8-page limit is a genuine challenge, and we recognize that we erred on the side of brevity in places. In the revision, we will add a more explicit discussion of why the martingale property is necessary.
>
> The key reference is Ramdas et al. (2023), "Admissible anytime-valid sequential inference must rely on nonnegative martingales," which formally establishes that martingales are not merely a convenient proof device but a necessary foundation for sequential inference. The pattern in our paper is: (1) establish that the estimation error is a martingale, (2) apply a strong invariance principle to approximate the sample paths by a Wiener process, and (3) apply Ville's inequality for nonnegative supermartingales to obtain a time-uniform bound on the magnitude of the estimation error. Without the martingale property, neither the strong approximation nor Ville's inequality applies, and one cannot guarantee coverage at all times simultaneously. We will make this reasoning explicit in the revision.
>
> **W2: why would [the union bound] be a bad strategy in the case of AIPW?**
> it would not be, and indeed our method uses event-time AIPW with the union bound. The union bound is applied to combine the two single-arm confidence sequences into a treatment effect confidence sequence (Theorem 4.9) (for both IPW and AIPW). Please see our note "Q3" to reviewer DzfE where we **significantly strengthen Theorem 4.4 to enable the AIPW estimator**.
> The conservatism of this union bound is already discussed in Proposition 4.12 and Remark 4.11, where we show it gives tighter intervals than the standard variance upper bound.
>
> **Q1: "Could you provide examples when you define the three proposed filtrations?"** We would be happy to include examples. In words:
> - Designer: We know everyone's assignment from entry, so future jumps are predictable — not a martingale.
> - Observer: We only learn $w_i$ at the observed event time, but the counterfactual estimator needs $w_i$ at the potential event time — not adapted.
> - Single-arm: We reveal $w_i$ at the potential event time $t_i(w)$, regardless of actual assignment — resolves both problems.
>
> **Q2: "Notation should be formally defined."** The notation $(\hat{\theta}_t \pm L_t)_t$ is simply used for the purposes of defining an asymptotic confidence sequence, following Waudby-Smith et al. (2024). These are generic placeholders, immediately instantiated in Theorem 4.6 with $\hat{\theta}_t = \hat{r}_t(w)$ and $L_t = b(\hat{V}_t(w); \alpha)$. We will add a sentence making this substitution explicit.

---

> > ### Author Rebuttal · Reviewer_KFyR · 2026-04-03
> >
> > Thank you for your detailed explanation. In veiw of the comments I am satisfied with the motivation of the model and significance of the results and I will raise my score accordingly.

---

### Decision · Program_Chairs · 2026-04-30

**Decision:**

Accept (regular)

**Comment:**

This paper solves the delayed-outcome causal inference problem in a design-based framework. Both the theoretical analysis and empirical results show the superiority of the proposed approach. An real application on OOM detection demonstrates the significant potential of the approach beyond theory and simulation study. The paper is technically sound and novel. After the discussion, all the reviewers give positive ratings and reach an agreement that this paper should be accepted.